# Embracing Diversity: Zero Shot Classification Beyond One Vector Per Class

## Abstract

Vision-language models for the first time enable open-world classification of objects without the need for any retraining. While this zero-shot paradigm marks a significant advance, even today's best models exhibit skewed performance when objects are dissimilar from their typical depiction. Real world objects such as pears appear in a variety of forms — from diced to whole, on a table or in a bowl — yet standard VLM classifiers map all instances of a class to a *single vector based on the class label*. We argue that to represent this rich diversity within a class, zero-shot classification should move beyond a single vector. We propose a method to encode and account for diversity within a class using inferred attributes, still in the zero-shot setting without retraining. We find our method consistently outperforms standard zero-shot classification over a large suite of datasets encompassing hierarchies, diverse object states, and real-world geographic diversity. We also find our method scales efficiently to a large number of attributes to account for diversity—leading to more accurate predictions for atypical instances. Finally, we highlight how our method offers fine-grained human-interpretable explanations of model predictions. We hope this work spurs further research into the promise of zero-shot classification beyond a single class vector for capturing diversity in the world.

## 1 Introduction

A pivotal advance in machine learning is the advent of *foundation models*. A single foundation model trained on large-scale data can supplant multiple task-specific models. Vision-Language models (VLMs) are popular foundation models capable of encoding text and images in the same representation space. Compared to standard classifiers which can only classify objects from a predefined list of classes, VLMs are capable of open-world, zero-shot classification—meaning, VLMs can classify any object using text descriptions without any additional training. This zero-shot paradigm has spurred the development of many VLMs Radford et al. (2021); Li et al. (2023); Yu et al. (2022) with impressive classification performance.

Despite their remarkable performance, even today's best models exhibit skewed performance for certain groups of images. For example, Richards et al. (2023) show models such as CLIP have exacerbated the gap in performance between regions such as Africa and Europe (as well as the gap across income-levels). We find similar biases arise when an object is visually dissimilar from its typical depiction. For example, Figure 1 (left) shows CLIP's $97.3\%$ accuracy on typical pears drops dramatically when a pear is peeled ($45.2\%$) or puréed ($30.3\%$). Addressing such biases is crucial to the reliability of classifiers in the real world, where instances within a class can vary significantly.

Zero-shot classifiers like standard models use a single vector in deep embedding space to describe an entire class. For standard zero-shot classification, a vision-language model (i) encodes the image along with 80 hand-crafted prompts per class name (e.g., "a photo of a pear" or "a drawing of a pear"), (ii) averages the 80 embeddings per class to obtain a single vector, (iii) predicts the class whose vector maximizes cosine similarity to the image embedding (Radford et al., 2021). Prompt averaging encourages all instances of a class to be mapped to the same vector in the model's embedding, inherently limiting the model's ability to infer the innumerable diversity within a class. A pear can be diced, sliced, whole, in one's hand, or in a bowl. In each case, the image of the pear would be markedly different, and its embedding may not always be well aligned with the single

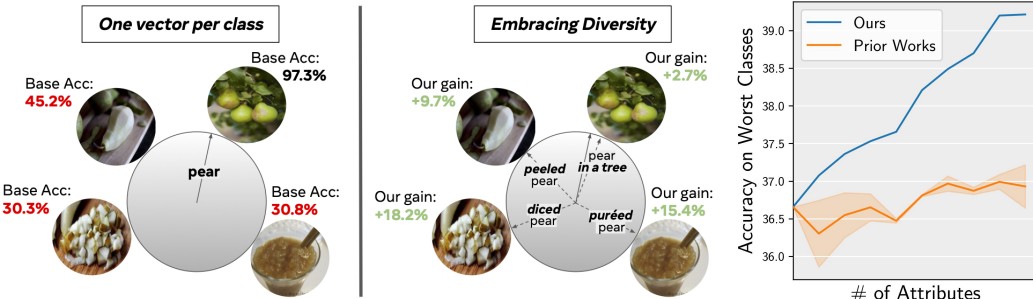

Figure 1: (**Left**) Instances of a class can appear in many diverse ways, like the pears above. Using one vector (the classname embedding) to represent the whole class results in disparate performance, particularly for atypical instances. (**Middle**) To address this issue, we infer attributes and embed multiple vectors. (**Right**) Our method scales better than prior works as we include more attributes, enabling us to account for the many ways in which diversity can arise. See Section 5.3 for more.

vector that is supposed to represent the *entire* class. Thus, there is a natural tension between the one vector per class paradigm and performing consistently across a class with high diversity, which we empirically validate.

While many strategies exist to mitigate performance disparities when labeled-data is available, these methods do not transfer to the data-free setting of zero-shot classification. Fortunately, unlike standard classifiers, the open-world nature of VLMs enables them to represent any attribute using the text encoder. VLMs can enrich the single per-class vector with attributes to more faithfully capture the variety with which a class can appear, pinpointing whether a pear is peeled or puréed. Thus, we argue that instead of learning one vector per class that is invariant to diversity, we should leverage the open-world nature of VLMs to *explicitly account for* the diversity within a class. Modern zero-shot classifiers warrant a modern paradigm: going beyond a single vector per class.

Recent work offer promising signs that zero-shot classification can be improved by incorporating attributes beyond the class name, such as subclasses (Novack et al., 2023) or visual descriptors (Menon & Vondrick, 2023; Pratt et al., 2023). However, the former is limited to datasets with hierarchical label sets, and the latter reverts back to the one vector per class paradigm via simple averaging, limiting the benefits of incorporating more attributes (Section 5.3). Importantly, diversity comes in many forms that generic descriptors or subclasses alone may not adequately capture.

In this work, we propose a zero-shot method for enriching classes with open-ended attributes to boost zero-shot classification. Our method consists of two steps: 1) an attribute inference step, in which we use generative language modeling (an inherent, under-utilized capability of modern VLMs) to enumerate relevant attributes along *many* various axes of diversity, and 2) a prediction consolidation step, where we flexibly attend only to subpopulations (i.e., instances within a class sharing an attribute) that are most relevant to the image. By enriching and carefully consolidating attributes to describe diversity within a class, our method more faithfully encodes atypical instances.

By going beyond a single vector per class we find consistent gains in zero-shot classification over a large suite of datasets encompassing hierarchies, diverse object states, and real-world geographic diversity. We find that our method consistently improves over standard zero-shot classification without any additional training. Encouragingly, we find gains often stem from better coverage for the hardest classes and subpopulations, where atypical instances are usually found. Our method additionally features enhanced interpretability, where each inference comes with the specific list of fine-grained attributes used to predict the class. Compared to existing methods, we find that our approach can effectively scale to a much larger number of attributes to cover broader axes of diversity as shown in the right panel of Figure 1. Our method also offers a principled trade-off between accuracy overall vs. on the worst classes, all without additional training. In summary, we (i) identify a limitation of the one-vector-per-class paradigm in adequately representing classes with diverse subpopulations, (ii) propose to go beyond one vector per class, leveraging under-utilized abilities of VLMs to explicitly account for intra-class diversity, and (iii) extensively validate the effectiveness of our method in improving performance, specifically for diverse subpopulations.

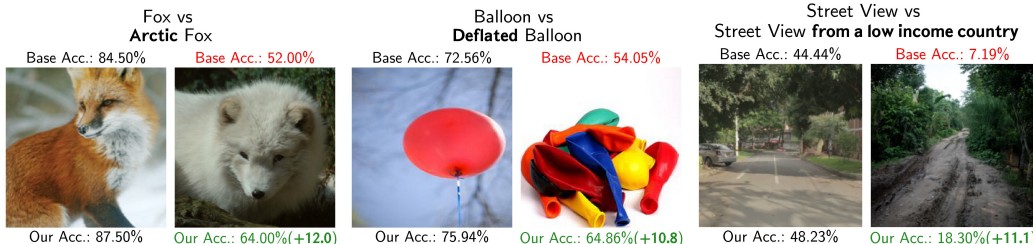

Figure 2: We test models on datasets that provide groundtruth **attributes** (shown in bold) annotating hierarchies, diverse states, and real-world shifts (e.g., Rojas et al. (2022) labels the income level and country of origin of each image, towards promoting AI models that reduce bias) within a class. We find that standard zero-shot accuracy ('Base Acc.' above) drops significantly when certain attributes are present, namely when the attribute manifests in visual differences from what the model considers 'typical' for the class. We design our method to improve performance on these 'atypical' instances.

## 2 REVIEW OF LITERATURE

Despite impressive overall accuracy, modern classifiers still suffer from biases. That is, they under-perform on some parts of the data, often due to spurious correlations or data imbalances in the training set. These biases can result in significant negative real-world impact. For example, Buolamwini & Gebru (2018) exposed significant bias along demographic lines for facial recognition systems, and more recently, Richards et al. (2023) demonstrated that despite steady progress on typical benchmarks, today's best models still generalize poorly to images from lower-income households and certain geographic regions. Namely, VLM-based zero-shot classifiers were shown to have even larger performance disparities across geographic and economic shifts than their supervised counterparts.

However, the promise of open-world zero-shot classification rightfully draws much attention to VLMs, which operate by mapping images and text to a shared latent space. CLIP (Radford et al., 2021), a seminal VLM, achieves this via joint contrastive training of image and text encoders on 400 million image-caption pairs. Recent models such as BLIP-2 (Li et al., 2023) bootstrap the training of more powerful VLMs by taking larger pretrained vision and language backbones and fusing their outputs to a single space, which in turn can even be used to generate text; that is, some modern VLMs contain a fully functional LLM with (often under-utilized) generative abilities. To perform zero-shot classification with VLMs, one computes the class that has the highest cosine similarity between a test image's embedding and the embedding of a class name, often averaged over many (80 for CLIP) handcrafted prompt templates. While many efforts have improved VLM-based classification via prompt-tuning Zhou et al. (2022b;a); Zhu et al. (2022); Derakhshani et al. (2023); Huang et al. (2022); Mirza et al. (2023); Menghini et al. (2023), nearly all require some labeled data. Other works focus more closely on the task of debiasing VLM-based classifiers Chuang et al. (2023); Seth et al. (2023); Zhang et al. (2023); Kim et al. (2023), though they too utilize labeled data, placing them out-of-scope of the true zero-shot setting.

Compared to previous classifiers, the key novelty of VLMs is their ability to encode *any* text. However, standard zero-shot classifiers only embed classnames, either alone or averaged over prompts. We propose to leverage the open-vocabulary capabilities of VLMs to improve coverage of intra-class diversity by embedding more than just the class name. One effort along these lines is Perception-CLIP (An et al., 2023), a concurrent work that infers contextual attributes per image as generative factors and does class inference conditioned on them. Other works utilize LLM-generated class descriptors, towards creating a concept-bottleneck (Yang et al., 2023) or rationales for inference (Feng et al., 2023), though these methods use data to train a linear layer atop descriptor similarities. DCLIP (Menon & Vondrick, 2023) show including descriptors can also improve performance in the zero-shot setting, and Pratt et al. (2023) extend the gains using additional handcrafted queries. WaffleCLIP (Roth et al., 2023) shows that appending random characters or words can achieve similar performance to descriptor-based methods like DCLIP, without the need for an external language model. Importantly, although these works obtain more than one vector per class, they all ultimately average over them. Thus, decision boundaries remain linear and biases may linger, as atypical instances are still suboptimally uncovered (see Sections 4.2 and 5.4). In contrast, like us, CHiLS (No-

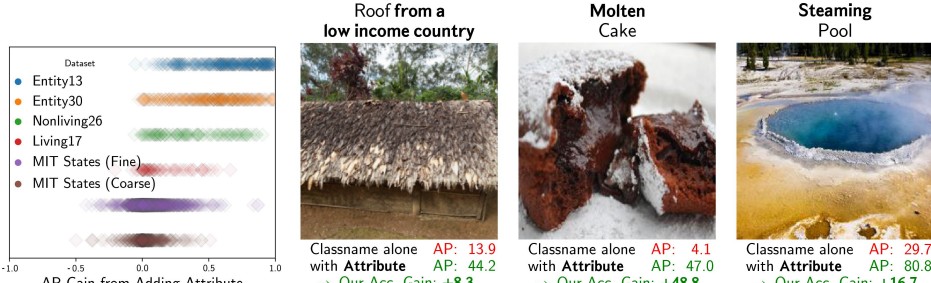

Figure 3: The average precision (AP) of a classname embedding is often much lower than the average precision of a subpopulation (i.e. classname with attribute) embedding. Subpopulations that see large increases in AP by including the attribute tend to be atypical. We design our method to improve accuracy on these diverse subpopulations, by inferring and explicitly accounting for them.

vack et al., 2023) introduces a *non-linearity* in three steps: they (i) define subclasses with groundtruth label hierarchies or by querying GPT-3, (ii) do zero-shot classification on this extended set of classes (subclasses) and original classes, (iii) reweight the standard zero-shot score for each class with the max score from step (ii) over subclasses within the class. However, CHiLS is designed specifically for hierarchical label sets, which limits the types of diversity it can capture (see Section 5.3).

## 3 MOTIVATION

We hypothesize that the standard one-vector-per-class paradigm poses a tension for highly diverse classes. We investigate this by measuring classification performance as a function of class diversity. Indeed, we find classes with higher diversity suffer worse performance under the one-vector-per-class classification paradigm. Then we illustrate how the newfound open-vocabulary capability of VLMs can enrich the single class vector to encompass diverse instances without additional training.

### 3.1 A SINGLE VECTOR INADEQUATELY REPRESENTS DIVERSE CLASSES

A standard VLM classifier is most effective when it aligns all instances of a class to their class vector (and away from vectors for other classes). Intuitively, aligning instances with high diversity is challenging as their image embeddings are more dispersed—and particularly tough for fixed open-vocabulary VLMs that do not benefit from knowing the specific classes of interest during their pre-training (see Appendix F.1). We see in Figure 2 for example the less typical *Arctic* `fox` is far harder to recognize than a typical `fox` (52.0% versus 84.5% accuracy). We observed similar drops in accuracy for a *deflated* `balloon` versus a regular `balloon` and an *unpaved* street versus a paved one. To systematically quantify this tension, both for VLMs and for the one vector per class paradigm generally, we examine class accuracies on ImageNet (Deng et al., 2009) relative to the diversity of each class across several models with varying levels of supervision. To proxy diversity, we measure the variance of image embeddings within a class. In all cases, we observe a strong negative correlation between class-wise accuracy and diversity (see Table 3 and details in Appendix C). That is, **classes with higher diversity have lower accuracy in the one vector per class paradigm**.

### 3.2 A PATH FORWARD: VLMS CAN RECOGNIZE DIVERSITY WITH RELEVANT ATTRIBUTES

Although standard VLMs use solely classname in zero-shot classification, their shared embedding space allows to encode relations to any other text. In turn we ask: *can the open-vocabulary encoder of VLMs better situate diverse classes given relevant attributes?* Specifically, we assess whether enriching classes with attributes can improve zero-shot classification on a suite of datasets with ground-truth attributes per class (details in Appendix B). We form a **subpopulation** by taking instances within a class that share an attribute. For each subpopulation, we compute the similarity of image embeddings with the text embedding of (i) the classname and (ii) the classname with the corresponding attribute, using CLIP ViT-B/16. We then measure the average precision of the two similarity scores for distinguishing instances within the subpopulation from instances outside of the

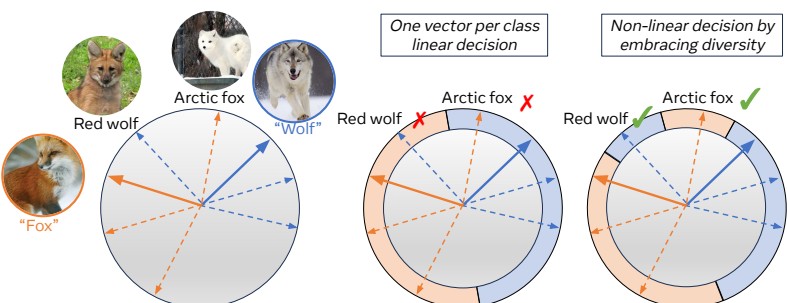

Figure 4: An *Arctic* `fox` can more closely resemble a typical `wolf` than a typical `fox`. Standard zero-shot classification using one vector per class (the classname embedding) is ill suited for this case. We address this issue by nonlinearly consolidating similarities to *multiple* vectors per class that explicitly encode the diverse subpopulations within the class. See section 4.2 for full explanation.

class. We find, as shown in Figure 3, that for the vast majority of cases, incorporating attributes leads to more precise recognition, and often by large margins: adding `molten` to `cake` improves average precision by over 40 points. Upon inspection, the highest gains in average precision tend to occur for atypical subpopulations (see Appendix B). Thus, VLMs *can* recognize instances in a class even when they are atypical, but this ability is restricted under the one vector per class paradigm.

## 4    METHOD

We now propose a method to better utilize the ability of VLMs to recognize diverse subpopulations. Our method consists of **attribute inference** and **prediction consolidation**. First, we query a large language model (LLM) for diverse per-class attributes that span many (often overlapping) subpopulations. Then, after computing the similarity of an image to each subpopulation, we non-linearly consolidate these similarities to obtain one score per class. We elaborate on these two steps below.

### 4.1    ATTRIBUTE INFERENCE ALONG MANY AXES OF DIVERSITY

To better cover the diverse subpopulations that may exist within a class, we incorporate attribute information. However, **diversity can come in many forms**. That is, the way in which two instances of a class differ can itself vary. Consider the examples in Figure 2. The *Arctic* `fox` case shows how a class can contain distinct finer-grained categories. In a related manner, the state or condition in which the class instance is in can also substantially change its appearance: a `balloon` looks much different when it is *deflated*. Further, there exist generic attributes that can lead to substantial visual differences regardless of the class, such as the region or income level of the country where an image is taken, exemplified by the two `Street View` images. Thus, to capture the many ways in which diversity can arise, we employ multiple distinct queries, in contrast to prior work. Namely, we infer:

- *Class specific* attributes, such as the possible **states** of an object (e.g., *diced* or *sliced* for `pear`). We also obtain **descriptions** for and different **kinds** of each class, as in DCLIP and CHiLS respectively.

- *Class adjacent* attributes, like **co-occurring objects** or **backgrounds**, to get useful context.

- *Class agnostic* attributes that describe how objects vary in general. For example, towards improving geographic fairness, we list potential choices for the **income-level, region** and **country** of origin of the image. We also introduce a novel two-step LLM query, where we first ask the LLM to list generic axes of diversity, and then have it populate those axes. We name this **auto-global** as it automatically generates many global attributes.

Appendix D.2 contains the exact LLM prompts and example inferred attributes for each query above.

### 4.2 Nonlinear Prediction Consolidation

Enumerating attributes along various axes of variation results in descriptions of many diverse subpopulations per class. Since VLMs have open-vocabulary text encoders, we can directly embed these subpopulation descriptions, in addition to the class name. Given a test image, we compute similarities to each of these embeddings. We then must consolidate them to obtain a single score per class.

Figure 4 illustrates the simple case of `fox` vs `wolf` classification, where solid/dotted lines correspond to classname/subpopulation embeddings on the hypersphere (shown here in 2D). The leftmost panel shows examples from the two classes near where their image embeddings would lie. Text embeddings for the subpopulations (dotted lines) are close to corresponding image embeddings, as VLMs are capable of recognizing even diverse subpopulations (see Section 3.2). Standard zero-shot inference maps a test-time image to the class of the nearest classname text embedding. Since there is only one vector per class (the classname-based embedding), the decision boundary is linear, as shown in the middle panel. The edge of the hypersphere is colored (orange for wolf, blue for fox) to indicate the predicted class for an image embedding at that location. Notably, the *Arctic* `fox` is misclassified as `wolf`, as its appearance more closely resembles a typical wolf than a typical `fox` and thus they fall closer to the text embedding of "wolf" (and vice-versa for the *red* `wolf`). Methods like DCLIP and WaffleCLIP embed more than just the classname, but they consolidate similarities via averaging, again resulting in a linear decision boundary. Even if atypical subpopulations are included at first, averaging can narrow the initial diverse coverage, as most embeddings for a class may better describe a typical instance.

In contrast, we propose the following *nonlinear* consolidation: we compute the single score per class for a given test image as the average of the similarities of the image embedding to *only the $k$ closest* subpopulations embeddings for the class, where $k$ is typically small (we use $k = 16$). This way, an image can have a high class score even if it is only similar to a small subset of subpopulations, as is the case for atypical instances. Thus, the *Arctic* `fox` and *red* `wolf` can be correctly classified despite being far from the classname and most subpopulation embeddings for their respective classes, as shown on the right panel of Figure 4, where we use $k = 1$ for simplicity (i.e. images are mapped to the class of the closest dotted or solid line, leading to a non-linear boundary). We shed insight on the effect of varying the hyperparameter $k$ in Section 5.4, revealing a tunable accuracy-fairness trade-off.

## 5 Analysis

We now empirically show that our method improves accuracy compared to strong baselines consistently across eight datasets, with larger gains for the hardest classes and subpopulations (which are likely more diverse and atypical, respectively). We then highlight (i) the enhanced interpretability of our method, (ii) the scalability of our method as more attributes are included, and (iii) an observed trade-off between accuracy overall and on the hardest classes that, notably, can be *controlled*.

### 5.1 Consistent Gains Across Diverse Datasets

We curate a suite of eight *attributed* (so to have groundtruth subpopulations) datasets spanning different axes of diversity. We use the four Breeds datasets (Santurkar et al., 2020) for their hierarchical label sets, as used in the CHiLS paper; indeed, these datasets were amongst those where CHiLS worked best. Next, we devise two classification tasks (coarse and fine grained) from the MIT States dataset (Isola et al., 2015) to track performance over labeled states (e.g., *sliced* or *diced* for `pear`). Importantly, we also include the datasets Dollarstreet (Rojas et al., 2022) and GeoDE (Ramaswamy et al., 2023), which contain images from varied geographic regions and income levels. As the diversity in these datasets is naturally occurring diversity, they can encompass *many* axes of variation, as opposed to our other datasets that only varying along known axes, like object state or kind.

We measure performance of zero-shot classifiers using the popular CLIP ViT-B/16 and BLIP-2 VLMs (Radford et al., 2021; Li et al., 2023). To infer attributes, we utilize the open source Vicuna-13b-v1.5 language model (Chiang et al., 2023), which notably is already contained in the BLIP-2 model we use. We report accuracy overall as well as averaged over the worst 20% of classes and subpopulations. Note that we only use groundtruth attributes when computing metrics; our method exclusively uses attributes *inferred* via the queries listed in Section 4.1. We also compute the lowest

| Dataset Type | | Accuracy | Avg Worst Subpop | Worst 20% of Classes | Worst 20% of Subpops |
|---|---|---|---|---|---|
| States | Vanilla | 66.71 | 40.66 | 35.46 | 21.73 |
| | DCLIP | 63.65 | 39.41 | 34.26 | 20.98 |
| | Waffle | 66.68 | 40.71 | 35.49 | 22.05 |
| | CHiLS | 66.56 | 40.41 | 36.16 | 22.45 |
| | Ours | **67.92** | **41.53** | **38.16** | **23.64** |
| Hierarchical | Vanilla | 78.15 | 48.36 | 50.72 | 35.89 |
| | DCLIP | 77.80 | 48.48 | 51.05 | 34.36 |
| | Waffle | 78.52 | 49.42 | 49.78 | 35.22 |
| | CHiLS | 79.44 | **52.65** | 51.80 | 38.44 |
| | Ours | **79.50** | 51.23 | **52.59** | **38.57** |

Table 1: Zero-shot classification on datasets with known variation types for CLIP with a ViT-B/16 encoder. States averages results over the two categorizations of MIT States data, while Hierarchical averages results over four Breeds datasets. We observe similar results for BLIP-2 (Table 7).

| *DollarStreet* Method | Accuracy | Worst Region | Worst Income | Avg Worst Subpop | Worst 20% of Classes | Worst 20% of Subpops |
|---|---|---|---|---|---|---|
| Vanilla | 51.51 | 42.43 | 34.76 | 37.60 | 18.33 | 11.01 |
| DCLIP | 49.78 | 41.08 | 32.91 | 36.37 | 19.07 | 11.19 |
| Waffle | 51.37 | 42.71 | 34.97 | 37.69 | 18.12 | 10.74 |
| CHiLS | 51.68 | 42.20 | 33.90 | 37.60 | 20.51 | 12.72 |
| Ours | **52.70** | **44.04** | **37.21** | **40.31** | **20.88** | **15.05** |
| *GeoDE* | | | | | | |
| Vanilla | 90.15 | 86.63 | - | 82.57 | 72.24 | 69.95 |
| DCLIP | 91.31 | 88.14 | - | 84.21 | 74.44 | 71.90 |
| Waffle | 91.59 | 89.06 | - | **85.44** | 75.85 | 74.37 |
| CHiLS | 90.96 | 87.90 | - | 84.48 | 73.27 | 71.64 |
| Ours | **91.75** | **89.12** | - | 85.40 | **76.13** | **74.64** |

Table 2: Zero-shot classification performance on geographically diverse images from DollarStreet and GeoDE using CLIP with a ViT-B/16 encoder. We observe similar results for BLIP-2 (Table 8).

subpopulation accuracy per class and average that, denoted as 'Avg Worst Subpop'. For the real-world shifts, we also report worst region and worst income group accuracy. Our baselines include: standard zero-shot (classname only) which we call Vanilla, DCLIP (averages over class descriptors), WaffleCLIP (averages over *random* descriptors sampled over ten trials), and CHiLS (reweights standard zero-shot class score with *max* probability of different kinds of the class). Notably, we average all text embeddings over the 80 prompts crafted for CLIP, so to report best possible baseline results.

Table 1 shows results for datasets with diversity along hierarchical and states axes, and table 2 shows results for geographic diversity. Our method consistently improves accuracy, even over CHiLS in the hierarchical setting it was specifically designed for. Notably, CHiLS becomes less effective for other datasets, while our method remains strong. We observe larger gains for worst class and subpopulation metrics, especially over baselines that consolidate via averaging, supporting the claim that our method improves coverage of the most atypical instances, and that moving beyond the one vector per class paradigm helps in this regard. For Dollarstreet, we see a $9\%$ average relative gain over baselines for the accuracy over worst income group metric, commonly used as a real-world fairness indicator, and an even larger gain for the worst $20\%$ of subpopulations.

## 5.2 FAITHFUL FINE-GRAINED INTERPRETATIONS FOR FREE

In addition to improving accuracy, our method has enhanced interpretability, as each inference comes with a list of the $k$ subpopulations specifically relevant to the test image for free. Figure 5 shows a few examples (see Appendix A for more). These interpretations are faithful, as they are exactly the subpopulations used to compute the class score. Also, since we include attributes along var-

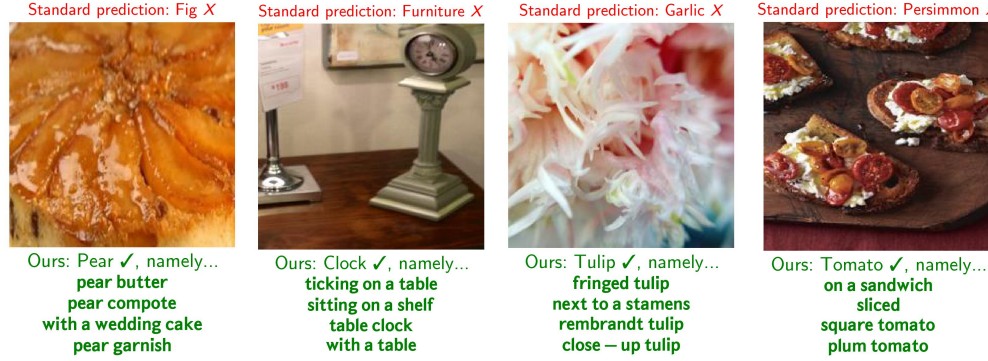

Figure 5: Atypical instances that our method can correctly recognize. The attributes used in inference also serve as fine-grained explanations, which can aide in model debugging.

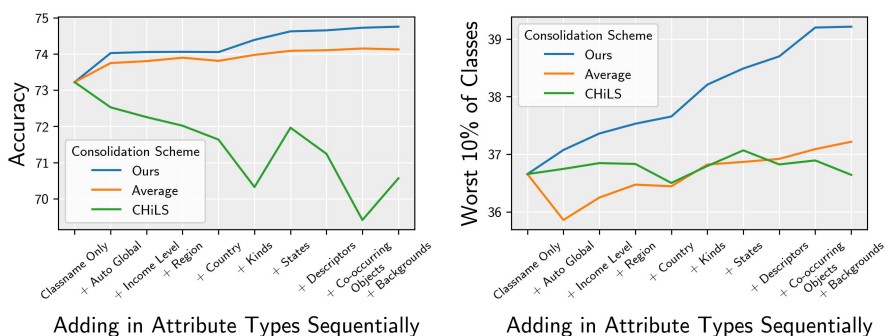

Figure 6: Accuracy, overall and for the worst classes, as new types of attributes are added. Performance for our consolidation scheme continuously improves, while it saturates or deteriorates for others. Figure 11 shows similar trends for accuracy on the worst 20% of classes and subpopulations.

ious axes of diversity, our interpretations are finer-grained than prior work: DCLIP yields the same set general descriptors for any image predicted to a given class; WaffleCLIP offers no interpretability. This interpretability can enable model debugging and facilitate increased trust with an end user.

## 5.3    SCALING WITH THE MANY AXES OF DIVERSITY

One source of gains for our method is that we infer attributes of *many* types, while prior works only include one. We argue that our flexible consolidation (of subpopulation similarities to a single class score) also provides improvements over naive averaging or the nonlinear consolidation of CHiLS. To test this, we sequentially add each type of attribute, and inspect performance using the three methods. Figure 6 shows our consolidation scales best as more attributes are added, with sizable gains for accuracy over the worst classes. In contrast, performance saturates with averaging, and actually deteriorates with CHiLS. The latter occurs since CHiLS assumes that subpopulations are mutually exclusive, as is the case in hierarchical label sets. When adding attributes along the many axes of diversity, resultant subpopulations overlap, making a zero-shot classification over all subpopulations (as done in CHiLS) unreliable. Averaging is also suboptimal, as the impact of each attribute diminishes as the number of attributes added increases: we see this in the left plot, as accuracy barely increases for the final three added attribute types. Also, samples that are close to only a few subpopulations but far from most (i.e., atypical instances) ultimately receive a low score when all scores are averaged. Thus, while averaging over subpopulations can improve accuracy (to an extent), it is less suited to improving performance on atypical instances than our method. We explore this further in the next section.

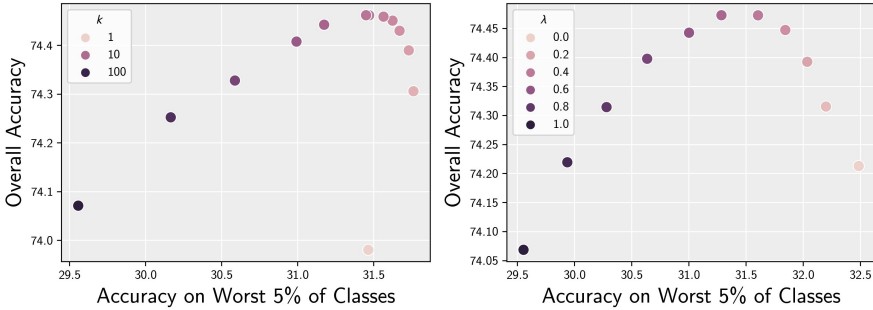

Figure 7: **Right:** As $k$ decreases, first, accuracy overall and on the worst classes both increase. Then, overall accuracy begins to decrease while accuracy on the worst classes continues to improve. Thus, we can control this trade-off via $k$. **Left:** $\lambda$, the continuous analog of $k$, allows for greater control.

## 5.4 Tunable Trade-off between Accuracy Overall and On Worst Classes

Recall that our method consists of computing the similarity of a given test image's embedding to the embedding of numerous (on the order of hundreds) subpopulations per class, before averaging over only the top $k$ similarities, where $k$ is small. Note that when $k = \infty$, our consolidation reduces to simple averaging over all vectors per class. To shed insight on how our consolidation differs from averaging, we sweep $k$, while keeping our attribute inference fixed. Additionally, we explore linearly interpolating class scores using our consolidation (top-$k$) and full averaging via a second hyperparameter $\lambda$, so that $\lambda = 0$ results in our method and $\lambda = 1$ is averaging. We jointly sweep $\lambda$ from $0$ to $1$ and $k$ from $1$ to $128$ to pinpoint the way in which our consolidation improves upon averaging.

Figure 7 shows overall accuracy vs. accuracy on the worst $5\%$ of classes[1] for both $k$ and $\lambda$. The trend is identical for the two parameters: first, both accuracy metrics increase as we move away from full averaging, with much larger gains occurring for the worst classes. Then, accuracy begins to drop, while accuracy on the worst classes continues to improve. To understand this trade-off, consider an instance that has high similarity to one subpopulation embedding for a class, and low similarity to all others. In the $k = 1$ case, this instance is given a high score for the class. This can benefit atypical instances of the class, as they may be visually dissimilar from most other instances (recall the *Arctic* fox). However, this can introduce errors, as the correct prediction for an instance mostly close to embeddings from its true class can be flipped with the presence of just one highly similar (perhaps unreliable) subpopulation embedding from a different class. Thus, lower choices of $k$ may benefit more atypical instances, leading to improved accuracy on worst classes (which are most diverse; see 3.1), potentially at the cost of overall accuracy. With this insight, practitioners can choose how to tune our method based on their end goals. Also, since $\lambda$ is continuous, it offers closer control of this tradeoff: indeed, accuracy on the worst classes can be improved by a larger margin when varying $\lambda$, and varying $k$ and $\lambda$ together can lead to best numbers for both metrics.[2]

## 6 Conclusion

To represent classes with diverse instances, which can come in many forms, one vector per class may not be enough. Further, VLMs have amazing abilities that are restricted when we only use one vector per class. Thus, instead of ignoring intra-class diversity, we *embrace* it, by explicitly inferring and encoding as much of it as we can. We propose a simple nonlinear consolidation scheme that flexibly attends to subpopulations present in an image while ignoring those that are irrelevant. We find that our method consistently improves over strong baselines, and careful ablations indicate that our method's gains come from improving performance on the hardest classes and subpopulations. We hope our work spurs further curiosity around how existing paradigms may limit the capabilities of our modern models, towards developing new paradigms to address problems with real world impact.

---

[1]We observe the same trade-off when inspecting the worst $10\%$ and $20\%$ of classes. See Appendix 11.

[2]To be true to the zero-shot setting, no tuning was done to obtain the results in 5.1. We tried two reasonably small values for $k$ (8 and 16), observed similar results, and went with $k = 16$, which was marginally better.

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

Standard prediction: Moss ✗

Standard prediction: Plate ✗

Standard prediction: Persimmon ✗

Ours: Pear ✓, namely...
*pear smoothie*
*pear juice*
*smooth pear*
*pear puree*

Ours: Pear ✓, namely...
*pear on a plate*
*pear in a bowl*
*pear in a fruit bowl*
*pear in a kitchen scene*

Ours: Tomato ✓, namely...
*cooked*
*in a pasta dish*
*sauced*
*roasted*

Standard prediction: Shower ✗

Standard prediction: Monkey ✗

Standard prediction: Monkey ✗

Ours: Balloon ✓, namely...
*as a tool for advertising*
*varying shades of red, yellow, or green*
*as a toy for a child*
*matte balloon*

Ours: Ape ✓, namely...
*next to a flower*
*red ape*
*showing affection*
*from indonesia*

Ours: Ape ✓, namely...
*showing affection*
*from sub − saharan africa*
*common chimpanzee ape*
*playing with others*

Standard prediction: Wolf ✗

Standard prediction: Chocolate ✗

Standard prediction: Well ✗

Ours: Fox ✓, namely...
*arctic fox*
*white fox*
*soft furred fox*
*curled up in a den*

Ours: Clock ✓, namely...
*in a pocket watch form*
*pocket watch clock*
*tiny clock*
*small clock*

Ours: Canyon ✓, namely...
*wood canyon*
*serene and peaceful*
*verdant and lush*
*next to a trees*

Figure 8: Our method yields faithful, fine-grained interpretations, for free. Top 4 shown for brevity.

## A  EXAMPLE INTERPRETABLE INFERENCES

We show additional examples of interpretable inferences in figure 8.

Figure 9: Example subpopulations where our method exhibits sizable accuracy gains compared to standard zero-shot classification (i.e. classname embedding only).

## B   CASES WHERE ATTRIBUTES HELP MOST

Figure 9 show more qualitative examples where standard zero-shot classification leads to biased performance. We highlight examples that our method leads to improvements. Notice that the subpopulations tend to be atypical.

Figure 10 shows more examples of subpopulations where including the groundtruth attribute results in significant gains in average precision. Again, these subpopulations generally appear differently than a typical instance from their class. Thus, the classname embedding is imprecise. However, evidently, VLMs are still capable of recognizing the subpopulation when given the attribute.

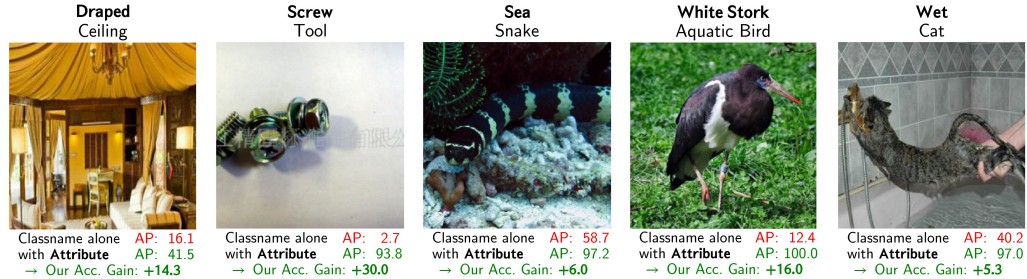

Figure 10: Example subpopulations where the classname embedding is imprecise, but including the attribute leads to large boosts in average precision. Notably, these subpopulations reflect instances atypical to the class.

| Classifier Encoder | CLIP | DINO | Sup. |
|---|---|---|---|
| CLIP | -0.28 | -0.51 | -0.43 |
| DINO | -0.37 | -0.54 | -0.48 |
| Sup. | -0.47 | -0.72 | -0.65 |

Table 3: Correlation between diversity and accuracy by class on ImageNet. We study three models: vision transformers trained with CLIP, DINO, or traditional label supervision. Diversity refers to variance of image embeddings within a class, with embeddings obtained with the 'encoder' model.

## C  DETAILS ON CORRELATION BETWEEN DIVERSITY AND ACCURACY PER CLASS

We compute ImageNet accuracy per class using three models: CLIP ViT-B/16 via standard zero-shot classification, DINO ViT-S/16 with a linear classification head fit to ImageNet over fixed features Caron et al. (2021), and a ViT-S/16 trained with traditional class-label supervision on ImageNet Touvron et al. (2021). Notably, all these models utilize a linear classification head. That is, they operate under a one vector one class paradigm. To proxy diversity, we measure the variance of embeddings per class. That is, per class, we compute the average squared distance between the mean embedding and the embedding of each class instance. Note that our measure of diversity depends on the image encoder; we explore using each of the three aforementioned models. Table 3 shows the results. All correlations are strongly negative, indicating that across classifiers and using various measures of diversity, classes with higher diversity are predicted at lower accuracies. This supports the intuitive hypothesis that consistently representing an entire class with one vector is made challenging when the class contains diverse instances.

## D  ADDITIONAL EXPERIMENTAL DETAILS

Note that we will provide all code, so that further details are easily accessible.

### D.1  DATASETS

The four hierarchical datasets we utilize are subsets of ImageNet Deng et al. (2009) curated by Santurkar et al. (2020). We also utilize the attributed dataset of MIT States Isola et al. (2015), deriving two classification tasks from their annotations. Finally, we utilize the geographic fairness benchmarks of Dollarstreet Rojas et al. (2022) and GeoDE Ramaswamy et al. (2023). When reporting subpopulation accuracies, we use income level as the ground truth attribute for Dollarstreet. Note that for MIT States and Dollarstreet, we conduct a filtering of classnames. Namely, we compute cosine similarity of CLIP embeddings for each pair of classnames. For any pair exceeding a threshold, we remove one classname from consideration. We do this because MIT States was not originally intended to be a classification dataset, and we observed highly similar classnames in Dollarstreet

| Query | Prompt | Examples |
|---|---|---|
| Kinds | List 16 different kinds of pear | Bartlett, Bosc, D'Anjou |
| States | List 10 different ways in which a pear may appear in an image | Whole pear, Pear slices, Pear chunks |
| Descriptors | List useful features for distinguishing a pear in an image | Round shape, Glossy skin, Green or brown color |
| Co-occurring Objects | In an image of a pear, list 10 other objects that may also appear | Leaves, Stem, Branches |
| Backgrounds | List ten different locations in which a pear may appear in an image | Fruit basket, Still life painting, Candy dish |

Table 4: Example LLM prompts and outputs for class-specific and class-adjacent queries.

(e.g. 'toilet' and 'bathroom/toilet'). We use a threshold of $0.8$ and $0.9$ to generate the coarse and fine-grained MIT States datasets respectively, and use a threshold of $0.9$ for Dollarstreet.

## D.2 INFERRING ATTRIBUTES

We now provide details on our exact LLM queries. First, for class-specific and class-adjacent queries, table 4 shows the precise prompt shown to the LLM along with example outputs, both for the class `pear`. For all queries, we append `Only use up to three words per list item` so that the LLM does not drone on. We sample from the LLM (Vicuna-13b-v1.5) with a temperature of $0.7$, repetition penalty of $1$, and a max number of new tokens of $512$.

We now provide more information on class-agnostic queries. We use continents as regions, and the five most populous countries per continent as our list of countries. These can both be obtained via prompting an LLM or searching the internet.

## D.3 AUTO GLOBAL

We now show more details for the auto-global query, which we found quite impressive. It consistently was amongst the attribute type that provided the most accuracy gains across datasets. The first prompt to the LLM was:

```
List 16 common general ways in which two instances of the
same object may look different.  For example, size, age, or
cleanliness.  Only use one word per list item.
```

The next prompt was:

```
For each of those items, list up to four different general
adjectives related to the time.  Please use common words..
```

Then, finally, out of laziness, we included a third prompt of:

```
Thanks.  Please organize your output as a python dictionary.
```

The resultant axes of variation and attributes per axis can be found in Table 5.

## E ADDITIONAL RESULTS

In the main text, we presented results using CLIP. Results for BLIP-2 can be found in Tables 7 and 8. Trends are consistent with results CLIP. For a global picture, we present results averaged over both VLMs and all datasets in table 6. Our method performs best over all metrics, again with largest gains occurring over the worst classes and subpopulations.

We also show results for each dataset individually in table 9. We find it encouraging that our results are consistent across both VLMs and for each of our eight datasets.

Further, for the analysis in Section 5.3, we show performance using the similar metrics of accuracy over the worst $20\%$ of classes and subpopulations, as shown in most tables. See figure 11. Trends are the same as in the main text, though slightly less pronounced. To be clear, our consolidation yields best performance, while others either saturate or deteriorate.

Lastly, we also show additional plots for the analysis in Section 5.4. In the main text, we plotted accuracy overall vs. over the worst $5\%$ of classes. We choose to show accuracy over the worst $5\%$

| Axis | Attributes | | | |
|---|---|---|---|---|
| size | small | medium | large | tiny |
| age | young | mature | ancient | old |
| cleanliness | dirty | clean | spotless | grimy |
| color | white | black | red | blue |
| texture | rough | smooth | soft | hard |
| material | plastic | metal | wood | fabric |
| shape | round | square | rectangular | triangular |
| position | upright | horizontal | vertical | diagonal |
| reflection | bright | dull | shiny | matte |
| transparency | clear | opaque | translucent | transparent |
| shine | glossy | matte | shiny | dull |
| pattern | striped | polka-dotted | plaid | solid |
| markings | spotted | striped | checked | speckled |
| surface | rough | smooth | bumpy | even |
| appearance | appealing | unappealing | attractive | unattractive |

Table 5: Attributes and axes of diversity inferred via the **auto-global** query. See D.3 for more information.

| Method | Accuracy | Avg Worst Subpop | Worst 20% of Classes | Worst 20% of Subpops | Worst 10% of Classes | Worst 10% of Subpops |
|---|---|---|---|---|---|---|
| Vanilla | 73.22 | 50.17 | 44.90 | 33.10 | 36.66 | 22.05 |
| DCLIP | 72.65 | 49.72 | 45.35 | 32.72 | 37.16 | 21.92 |
| Waffle | 73.36 | 50.23 | 44.97 | 33.34 | 36.66 | 22.43 |
| CHiLS | 74.13 | 51.84 | 46.00 | 34.80 | 37.07 | 23.24 |
| Ours | **74.75** | **52.04** | **47.52** | **35.77** | **39.21** | **24.40** |

Table 6: Average performance over eight datasets and two VLMs.

| Dataset Type | | Accuracy | Avg Worst Subpop | Worst 20% of Classes | Worst 20% of Subpops |
|---|---|---|---|---|---|
| States | Vanilla | 70.60 | 42.65 | 43.44 | 26.28 |
| | DCLIP | 69.80 | 41.42 | 41.54 | 24.25 |
| | Waffle | 70.10 | 42.18 | 41.99 | 25.76 |
| | CHiLS | 70.83 | 42.51 | **44.31** | 26.75 |
| | Ours | **71.30** | **42.84** | 43.92 | **27.21** |
| Hierarchical | Vanilla | 75.29 | 50.33 | 44.30 | 32.18 |
| | DCLIP | 75.60 | 49.41 | 46.35 | 32.25 |
| | Waffle | 75.25 | 48.84 | 44.48 | 31.67 |
| | CHiLS | 77.17 | 52.00 | 45.86 | 34.59 |
| | Ours | **77.95** | **52.47** | **48.66** | **35.46** |

Table 7: Zero-shot classification on datasets with known variation types for BLIP-2. Hierarchical datasets from Novack et al. (2023) and States are the average of coarse and fine-grained categorizations of MIT States. See table 1 for results using CLIP ViT-B/16.

| *DollarStreet* Method | Accuracy | Worst Region | Worst Income | Avg Worst Subpop | Worst 20% of Classes | Worst 20% of Subpops |
|---|---|---|---|---|---|---|
| Vanilla | 50.91 | 39.76 | 31.89 | 36.76 | 18.87 | 11.33 |
| DCLIP | 49.81 | 39.05 | 32.03 | 37.01 | 18.22 | 12.14 |
| Waffle | 51.07 | **41.00** | **33.05** | 36.67 | 19.43 | 12.53 |
| CHiLS | 51.56 | 40.26 | 32.37 | **38.35** | 19.56 | 12.45 |
| Ours | **51.96** | 40.63 | 32.78 | 37.91 | **21.04** | **13.61** |
| *GeoDE* | | | | | | |
| Vanilla | 90.48 | 87.95 | - | 84.41 | 71.01 | 69.06 |
| DCLIP | 90.98 | 88.19 | - | 84.78 | 72.71 | 71.32 |
| Waffle | 91.10 | 88.85 | - | 84.97 | **74.11** | **72.56** |
| CHiLS | 90.75 | 87.99 | - | 84.63 | 71.11 | 69.46 |
| Ours | **91.40** | **89.07** | - | **85.44** | 73.08 | 71.22 |

Table 8: Zero-shot classification performance on geographically diverse household object from DollarStreet and GeoDE using BLIP-2. See table 2 for results with CLIP ViT-B/16.

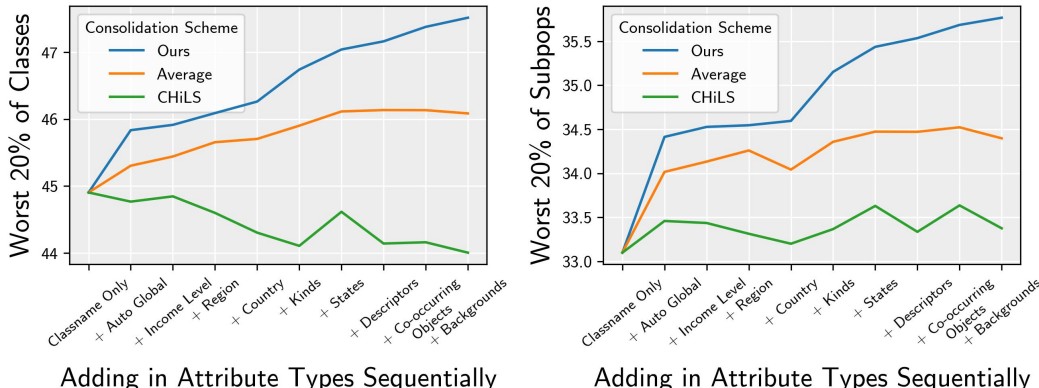

Figure 11: Accuracy for the worst 20% of classes and subpops, averaged over our dataset suite as we sequentially add new types of attributes using different consolidation schemes. See figure 6 in the main text for accuracy overall and over the worst 10% of classes, along with more discussion. As shown in the main text, our method scales the best as attributes are added sequentially.

| Method | Geographic | | (MIT) States | | Hierarchical | | | |
|--------|------------|-------|--------|------|----------|----------|-------------|----------|
| | Dollarstreet | Geode | Coarse | Fine | Entity13 | Entity30 | Nonliving26 | Living17 |
| Accuracy | | | | | | | | |
| Vanilla | 51.21 | 90.34 | 78.24 | 59.07 | 68.22 | 68.43 | 77.27 | 92.96 |
| DCLIP | 49.80 | 91.14 | 77.80 | 55.65 | 68.64 | 68.49 | 76.32 | 93.35 |
| Waffle | 51.22 | 91.34 | 78.31 | 58.47 | 68.95 | 68.66 | 77.13 | 92.80 |
| CHiLS | 51.62 | 90.85 | 78.83 | 58.55 | 69.33 | **70.69** | **79.55** | **93.65** |
| Ours | **52.33** | **91.58** | **79.33** | **59.90** | **71.47** | 70.59 | 79.25 | 93.59 |
| Average Worst Subpopulation Accuracy | | | | | | | | |
| Vanilla | 37.18 | 83.49 | 53.83 | 29.49 | 21.77 | 36.27 | 57.12 | 82.24 |
| DCLIP | 36.69 | 84.50 | 53.01 | 27.81 | 22.54 | 36.87 | 54.54 | 81.82 |
| Waffle | 37.18 | 85.20 | 53.94 | 28.96 | 21.88 | 36.87 | 56.37 | 81.41 |
| CHiLS | 37.98 | 84.56 | 53.69 | 29.22 | 23.77 | **42.50** | 59.50 | **83.53** |
| Ours | **39.11** | **85.42** | **54.26** | **30.10** | **25.31** | 39.03 | **59.54** | 83.53 |
| Accuracy for Worst 20% of Classes | | | | | | | | |
| Vanilla | 18.60 | 71.63 | 52.12 | 26.79 | 34.38 | 32.50 | 49.15 | 74.00 |
| DCLIP | 18.64 | 73.57 | 51.35 | 24.46 | 36.48 | 33.21 | 46.60 | **78.50** |
| Waffle | 18.78 | **74.98** | 52.31 | 25.17 | 31.41 | 33.46 | 48.40 | 75.24 |
| CHiLS | 20.04 | 72.19 | 53.65 | 26.82 | 36.07 | 31.71 | 52.05 | 75.50 |
| Ours | **20.96** | 74.61 | **54.03** | **28.05** | **37.55** | **34.94** | **53.10** | 76.92 |
| Accuracy for Worst 10% of Classes | | | | | | | | |
| Vanilla | 11.92 | 59.30 | 41.63 | 18.09 | 29.75 | 21.75 | 40.58 | 70.25 |
| DCLIP | 11.82 | 64.22 | 41.16 | 15.90 | 26.80 | 22.71 | 38.08 | **76.62** |
| Waffle | 10.69 | **64.74** | 42.00 | 16.68 | 23.03 | 23.79 | 39.78 | 72.59 |
| CHiLS | 13.64 | 58.82 | **44.74** | 18.41 | 25.60 | 21.04 | 42.92 | 71.38 |
| Ours | **14.35** | 62.61 | 44.24 | **19.29** | **31.10** | **25.33** | **43.50** | 73.25 |
| Accuracy for Worst 20% of Subpopulations | | | | | | | | |
| Vanilla | 11.17 | 69.50 | 36.23 | 11.78 | 14.54 | 15.62 | 33.90 | 72.07 |
| DCLIP | 11.67 | 71.61 | 35.08 | 10.16 | 14.54 | 14.92 | 30.19 | 73.57 |
| Waffle | 11.64 | **73.47** | 36.89 | 10.93 | 13.24 | 16.27 | 32.77 | 71.49 |
| CHiLS | 12.58 | 70.55 | 37.44 | 11.76 | 15.23 | 16.88 | **39.67** | 74.29 |
| Ours | **14.33** | 72.93 | **38.21** | **12.64** | **17.33** | 16.94 | 38.86 | **74.93** |
| Accuracy for Worst 10% of Subpopulations | | | | | | | | |
| Vanilla | 6.10 | 57.47 | 23.27 | 4.95 | 5.35 | 5.67 | 18.90 | 54.71 |
| DCLIP | 6.08 | 61.38 | 21.71 | 3.74 | 5.77 | 5.04 | 15.20 | 56.43 |
| Waffle | 5.82 | **63.27** | 23.27 | 4.26 | 4.96 | 6.56 | 17.00 | 54.30 |
| CHiLS | 7.40 | 57.62 | **24.93** | 4.88 | 5.96 | 6.21 | 21.50 | 57.43 |
| Ours | **8.62** | 61.26 | 24.72 | **5.53** | **6.88** | **6.88** | **22.00** | **59.29** |

Table 9: Metrics for each dataset. Results are averaged over CLIP and BLIP-2. Our method's gains are consistent over the eight dataset suite.

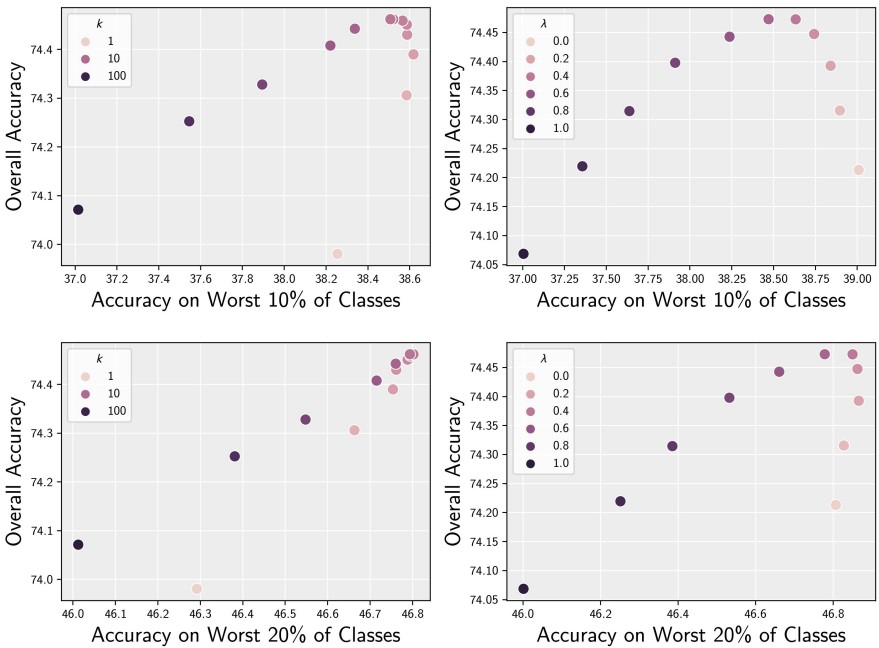

Figure 12: We replicate figure 7 using metrics that look at a larger portion of the worst classes. A similar tradeoff emerges, though in a slightly less pronounced way. We note that this is expected, as increasing the number of classes considered likely also increases the number of less diverse classes included.

because it most clearly conveys the tradeoff we observe. Figure 12 shows this tradeoff still exists when looking at other percentiles, though it is less pronounced, which is expected.

## F    WHEN CAN WE CRAM AN ENTIRE CLASS IN ONE VECTOR, AND WHEN CAN WE NOT?

Arguably, diversity within classes is unavoidable, as two instances can vary in numerous ways (discussed further in Section 4.1). How then, have classifiers enjoyed success under the one-vector-per-class paradigm, despite its tension with intra-class diversity? First, we note these performance disparities are often obfuscated in metrics like overall accuracy; indeed, the supervised classifiers studied above each achieve impressive overall accuracies. Nonetheless, the tension can be somewhat resolved if (i) one learns embeddings that reduce the diversity that is present in input space, and/or (ii) the single vector learned per class contains features that are unique to the class and present across class instances, despite intra-class variance that persists in the embedding space. We expand on these below.

### F.1    IDEAL CONDITIONS FOR THE ONE-VECTOR-ONE-CLASS PARADIGM

Most modern vision classifiers consist of a deep feature encoder, mapping images to a rich embedding space, followed by a linear classification head, mapping embeddings to class logits. The linear classification head consists of a single vector (and a scalar bias) per class. A linear classification head is accurate if, for any instance from the $i^{th}$ class, the activation on the $i^{th}$ class vector must be higher than the activation for any other class vector. We express this mathematically below, with $\mathbf{x}$ denoting the embedding of an image from class $i$, and $\mathbf{c_i}, \mathbf{c_j}$ denoting vectors in the classification head.

$$\forall \mathbf{x} \in \mathcal{C}_i, \forall j \neq i, \text{ we require that } \mathbf{x} \cdot \mathbf{c_i} - \mathbf{x} \cdot \mathbf{c_j} > 0 \tag{1}$$

$$\text{Note that } \mathbf{x} \cdot \mathbf{c_i} - \mathbf{x} = \mathbf{x} \cdot (\mathbf{c_i} - \mathbf{c_j}) = \|\mathbf{x}\|\|\mathbf{c_i} - \mathbf{c_j}\| \cos(\mathbf{x}, \mathbf{c_i} - \mathbf{c_j}) \tag{2}$$

$$\text{Thus, } \forall \mathbf{x} \in \mathcal{C}_i, \forall j \neq i, \text{ we require that } \cos(\mathbf{x}, \mathbf{c_i} - \mathbf{c_j}) > 0 \tag{3}$$

The last step arises because norm is always non-negative. Now, let us focus on different components of this required condition (by definition) for an accurate one-vector-per-class classification head. First, the single vector $\mathbf{c_i}$ must contain contain a set of features that are *unique* to that class. That is, these features remain when considering the residual $\mathbf{c_i} - \mathbf{c_j}$ for any $i \neq j$. Secondly, the unique features that discriminate the class from all others must also be aligned with every instance of the class. In other terms, these unique features must be *invariant* to any diversity within the class. Also, note that the quantity we expand upon above is simply the margin for classification. In the ideal case, this margin would be maximized.

## F.2 CLASS-SUPERVISED TRAINING IS WELL SUITED FOR THE ONE VECTOR PER CLASS PARADIGM, BUT VLM PRETRAINING IS NOT

In traditional class-label supervised training, the feature encoder is jointly optimized with the classification head to minimize a classification loss. Let us consider how this effects the linear classification head and the feature encoder individually. First, fixing the classification head, we see the supervised objective encourages all embeddings from one class to be drawn close to their respective single vector, and consequently, close to one another. In other words, invariance of embeddings within a class is promoted. Next, with the feature encoder fixed, classification head vectors align with embeddings within their class and de-align with embeddings from outside their class. Thus, the classification head vectors are optimized to solely contain the features unique to their class embeddings. Therefore, training with traditional class-label supervision directly promotes the invariance and uniqueness properties required for the success of the one-vector-per-class paradigm.

On the other hand, VLMs are optimized with markedly different objectives. Many VLMs employ contrastive image-text matching, in which negative examples are far weaker and classes are no longer defined; in some ways, the training is analogous to optimizing a classification task with an infinite number of classes. Indeed, two instances that belong to the same class in a downstream task may have embeddings pushed apart during VLM pretraining, directly going against the aforementioned notion of class-wise invariance. Other common VLM objectives like captioning or question answering promote the descriptiveness of the embedding. Thus, instead of honing in on unique features, embeddings are likely to describe as much as possible. We note that having maximally descriptive embeddings is typically a good thing, as it allows for re-use of the same feature encoder for many downstream tasks, as is done in linear probing with self-supervised encoders. The key caveat is that in those cases, the linear classification head is still exposed to instances from all classes, and thus, each classification head vector can learn to align only with the unique features for its class. In contrast, in the zero-shot setting, the classification head vectors are obtained independently of one another via embedding the names of classes via the text encoder, and thus, it is unreasonable to expect that these vectors satisfy the uniqueness condition.

## F.3 ARCTIC FOX CASE STUDY: BIAS CAN BE AMPLIFIED WHEN USING ONE VECTOR PER CLASS PARADIGM FOR ZERO-SHOT CLASSIFICATION

Staying in the one-vector-per-class setting, we now compare class vectors obtained directly in a zero-shot manner to those obtained with supervision. Specifically, we focus on the `Arctic Fox` bias, shown in Figure 2. We train a linear classification head over fixed CLIP embeddings used a skewed training set that under-represents `Arctic foxes` in the training set. We find that the bias of the zero-shot vector is on par with having only $3\%$ of the training images in the `fox` class be `Arctic foxes` in the supervised setting, suggesting that limitations of the one vector per class paradigm may be exacerbated in the zero-shot setting.

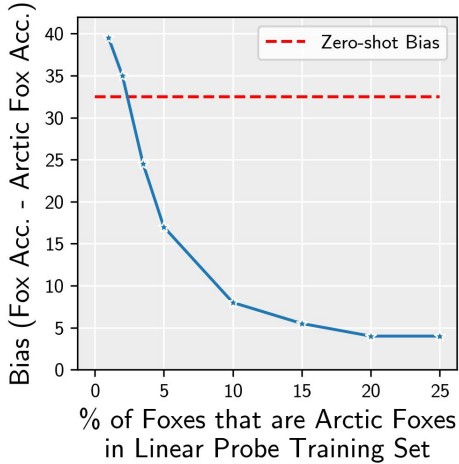

Figure 13: Arctic Fox bias is amplified in zero-shot classifier vs. to supervised linear probes.

## G  ONE FINAL TRADE-OFF

In section 5.4, we should two hyperparameters that could trade overall accuracy for accuracy over the worst classes. We now present one more, along with a *theoretical* explanation. Throughout the paper, we consider 'averaging' to mean computing similarities to multiple vectors and then averaging those similarities; this is how DCLIP and WaffleCLIP average, and will refer to this as **Average Sims**. However, averaging over prompts as done in originally in CLIP consists of averaging vectors first and then computing similarity to one average vector; we call this **Average Vecs**. The difference is subtle: in the latter case, an additional normalization occurs when cosine similarity is taken.

We now show theoretically that when all embeddings are normalized (i.e. for CLIP), **Average Vecs** simply rescales the class score yielded by **Average Sims** by a factor that measures how *diffuse* the vectors for the class are. Let $x$ be an image embedding and $\{v_1, v_2, \ldots, v_k\}$ be subpopulation vectors for a given class. We assume all vectors are normalized to the hypersphere, as is the case for CLIP. That is, $\|v_i\| = 1$ for all $i$ and $\|x\| = 1$. Let $\overline{v} := \frac{1}{k} \sum_{i=1}^{k} v_i$ denote the average vector. We compute the class score for **Average Vecs** below.

$$\textbf{Average Vecs} = \cos(x, \overline{v}) = \frac{x \cdot \overline{v}}{\|x\|\|\overline{v}\|} = \frac{x \cdot \frac{1}{k}\sum_{i=1}^{k} v_i}{\|\overline{v}\|} = \frac{\frac{1}{k}\sum_{i=1}^{k} x \cdot v_i}{\|\overline{v}\|}$$

$$= \frac{\frac{1}{k}\sum_{i=1}^{k} \cos(x, v_i)}{\|\overline{v}\|} = \frac{\textbf{Average Sims}}{\|\overline{v}\|}$$

To get from line 1 to 2, we utilize the fact that cosine similarity is equivalent to the dot product when both arguments are unit norm. Let us now consider what this result entails. The denominator is the norm of the average vector. This quantity is always between $0$ and $1$. It is lowest when the vectors are most diffuse. Thus, the class score obtained by **Average Sims** is scaled up to obtain the score for **Average Vecs** by more when the vectors are diffuse. In other words, averaging the vectors first implicitly upweights vectors corresponding to diverse subpopulations.

Based on this simple theory, we would expect the most classes with high diversity to have higher accuracy under **Average Vecs** compared to **Average Sims**, as their class scores are inflated more than the less diverse classes. The effect on overall accuracy, however, is not perfectly clear. To inspect this, we perform the same sweep over $k$ and $\lambda$ as in section 5.4, except now we additionally try replacing all similarity averaging with vector averaging. Figure 14 shows the results. We average away $k$ for clarity. Indeed, averaging over vectors improves accuracy on the worst classes. For high values of $\lambda = 1$, we see averaging vectors also slightly improves overall accuracy. However, in the vast majority of values for $\lambda$, overall accuracy is hurt by averaging vectors instead of similarities.

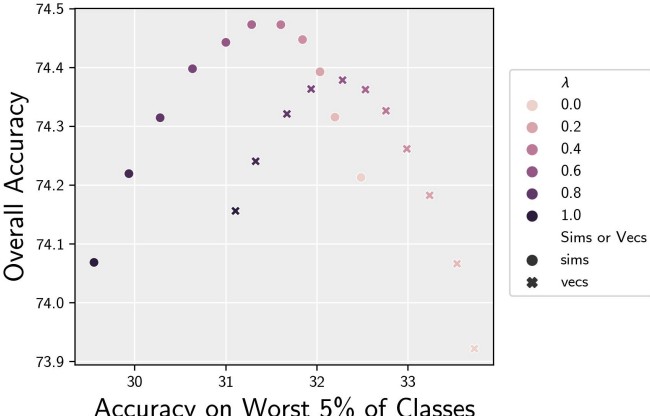

Figure 14: Averaging subpopulation vectors before computing similarity to an image embedding proves to be another way to trade overall accuracy for accuracy on the worst classes. That is, when we first compute similarity to each subpopulation and then average, we obtain higher overall accuracy but lower accuracy on the worst classes, compared to when we first average subpopulation vectors and then compute the similarity to the average vector.

We hope this analysis provides insight as to the precise effect of averaging similarities or vectors, which may be relevant to others who wish to explore going beyond one vector per class.

