# OpenReview forum: "Embracing Diversity: Zero-shot Classification Beyond a Single Vector per Class"
_ICLR.cc/2024/Conference — Submitted to ICLR 2024_

### Official Review · Reviewer_3g4D · 2023-11-01

**Soundness:** 2 fair
**Presentation:** 3 good
**Contribution:** 2 fair
**Rating:** 5
**Confidence:** 4

**Summary:**

This work studies the zero-shot classification problem. Rather than using a single vector to represent each class label, this work proposes to represent the rich diversity within each class using inferred attributes without any training. The proposed method is shown to outperform zero-shot classification methods (including DCLIP, Waffle, and CHiLS) on various datasets that contain hierarchies, diverse object states, and real-world geographic diversity (such as MIT States, Breeds, DollarStreet, and GeoDE).

**Strengths:**

The proposed idea of using VLMs to inferred attributes for zero-shot learning is valid, and it seems effective to use multiple attribute vectors per class in the zero-shot classification benchmark.

Using attributes can help to improve interpretability of the zero-shot inference results.

**Weaknesses:**

Even though using attributes is a valid idea in zero-shot learning/classification. The proposed method is not convincing. VLMs (such as CLIP) already has the zero-shot recognition ability, therefore, it seems a redundant inference step to use them for inferring attributes first and then for predicting the corresponding class labels. Why not directly applying the VLMs (e.g., CLIP) for zero-shot recognition? What are the empirical results using single-vector for zero-shot inference using CLIP or OpenCLIP.

The proposed method is also computationally more expensive compared to zero-shot inference with one vector. The compute requirement scales linearly to the number of attributes. Does the model performance improve and scale in proportion to the number of attributes? If not, why should one consider to add more compute for a more complicated inference process with not guarantee on performance improvement?

**Questions:**

Why not directly applying the VLMs (e.g., CLIP) for zero-shot recognition? What are the empirical results using single-vector for inference using CLIP or OpenCLIP.

Does the model performance improve and scale in proportion to the number of attributes?

**Details Of Ethics Concerns:**

No concern on Ethics.

---

> ### Author Response · Authors · 2023-11-17
> **Substantial gains over directly applying the VLMs; Low cost of our method**
>
> We thank the reviewer for their feedback. We now address the raised concerns.
>
> **Strong and consistent gains compared to using a single vector for zero-shot inference**: We note that we did indeed consider using the zero-shot capabilities of VLMs directly without attributes. This corresponds to our ‘vanilla’ baseline, where a single vector (i.e. the classname embedding) is used per class. **Our method significantly outperforms this baseline, with average (over 8 datasets and two VLMs) improvements of +1.5% in accuracy, +2.6% for the worst 20% of classes, and +2.7% for the worst 20% of subpopulations** (see table 6). Notably, we achieve higher accuracy than this vanilla baseline for every setting. In additional experiments over 9 new datasets conducted during the rebuttal period, we see similar gains over this baseline: +1.8% in accuracy and +1.5% for the worst 20% of classes* (see table A in the response to Reviewer KwVV). In fact, we see similar gains to stronger, more recent baselines that also incorporate additional information beyond classnames: compared to each baseline, on average, our method yields gains of 1.3% and 1.7% in accuracy overall and for the worst 20% of classes respectively.
>
> The reason why we infer attributes is to enrich the model’s coverage of *all* instances within a class, as the classname embedding often inadequately covers atypical subpopulations. The larger gains for worst class and subpopulation metrics suggest our method indeed improves on the baseline by better covering challenging (usually atypical; see sec 3.1) instances, showing that including attributes is well-motivated and effective. **Our method also provides clear interpretability advantages**; both the standard method and most baselines have no or little interpretability, while we offer fine-grained and faithful interpretations for each inference.
>
> *The new datasets do not have attribute annotations, so we cannot report accuracy on the worst 20% of subpopulations, as a subpopulation consists of all instances within a class that share an attribute.
>
> **On computational cost**: Crucially, **we only infer attributes and embed subpopulation vectors once for any classification task**. The other extra computations are all very simple (e.g. computing the similarity to more vectors only involves the dot product), and importantly, they take far less time than encoding the test image, which is a necessary step for any zero-shot method. Thus, aside from a constant cost done once and a small number of very quick extra computations, **our method is nearly just as fast as the standard method** *per test image*. Moreover, our method improves accuracy, performance on atypical samples, and interpretability. Finally, we note that **our method does indeed improve as more attributes are incorporated**, even though the cost of adding attributes is minimal. Further, our method is unique in this regard: Sec. 5.4 shows how **baselines saturate or deteriorate as more attributes are added, while ours continues to improve**. Thus, (i) the added computational cost is very small (and zero asymptotically), and (ii) our method continues to improve with more attributes, so we believe the benefits of our method certainly outweigh the very small extra cost.
>
> To summarize, we’ve studied the requested baseline, in the original submission and once more in a new set of evaluations during the rebuttal. In both cases, we see significant gains from our method. Also, we’ve explained how the added computational cost of our method is minimal. We hope these address your concerns, an if you feel as though they have, we’d greatly appreciate it if you could raise your score to an accept. We’d also be more than happy to answer any follow up questions if concerns linger. Thank you!

---

> > ### Author Response · Authors · 2023-11-21
> >
> > Hi, just a friendly reminder about our rebuttal above.
> >
> > To summarize the above post, we show **strong gains over the baseline you recommended**, including for **9 new datasets**, and explained that **our method is virtually as fast as other approaches**.
> >
> > We'd be very grateful if you could consider our responses before the **discussion period ends tomorrow**; thanks for the time you've spent already!

---

> ### Comment · Reviewer_NLHE · 2023-12-04
> **Final comments**
>
> First, I don't agree with R1 that there is no value to be able to have fine-grained zero-shot classification. In real-world, fine-grained classification is important, and it can be crucial to policy making if a model can correctly identify "arctic fox" from "wolves". This is especially true in conservation areas. We can not always rely on supervised learning approaches as it is not scalable. Therefore, I think it is important for people to keep looking for solutions for zero-shot fine-grained recognition in large vlm era. I think the comments from the author answered most of my concerns, and the method proposed is simple but still effective. However, like R1 mentioned, the performance is relatively limited, so I will keep my original rating. Thanks and sorry for the late response.

---

### Official Review · Reviewer_NLHE · 2023-11-01

**Soundness:** 3 good
**Presentation:** 3 good
**Contribution:** 3 good
**Rating:** 6
**Confidence:** 3

**Summary:**

This paper further explores VLM's zero-shot capacity by introducing non-linear hyperplanes, specifically through k-nearest neighbors. The diverse neighbors are achieved by using attributes of sub-classes within each class. The idea of employing sub-classes to enhance the variance of decision boundaries aligns well with the nature of VLMs, especially considering that VLMs typically consist of LLMs with open word space. The reported results also demonstrate the improvements introduced by the proposed method.

**Strengths:**

I think using diverse word attributes rather than limited words to represent recognition categories is a good idea. It well aligns with VLMs, showcasing the flexibility of VLMs compared to traditional one-vector based recognition protocol. The intuition why the author chose this route to address zero-shot with VLMs is clearly stated. The experiments also shows the validity of the method.

**Weaknesses:**

1) I think figure 4 is misleading. The idea is by using subclasses, the majority of close subclasses should be from the correct major class (correct me if I am wrong).  However, this figure does not show the two atypical classes have more close subclasses that make the two classes be classified to the correct class.
2) I think the proposed method may not work on fine-grained classes, as the variance of each class gets smaller and smaller.
3) The preparation of subclasses for each class may require even more effort than preparing hierarchical datasets or traditional attribute learning datasets.

**Questions:**

As above.

---

> ### Author Response · Authors · 2023-11-17
> **Clarifying fig 4, New results showing gains of our method on fine grained datasets, and Low cost of inferring attributes**
>
> Thank you very much for your feedback. We especially appreciate that you note how our method takes advantage of the complementary nature of LLMs and VLMs to tackle the problem of improving classification in the face of high intra-class diversity. We now address each of your concerns. We highlight that we conduct two additional experiments: (1) a large evaluation of 9 new datasets, (2) a time analysis of the added cost of our method.
>
> 1. Your intuition is very close to correct. One added detail, as shown in figure 4, is that our consolidation method allows for an atypical subpopulation to be classified correctly even when it is far from most subpopulation embeddings for that class. This property is unique to our consolidation method: using a single vector (either by not including attributes or by averaging over attributes), a sample is penalized if it is far from most subpopulations, even if the subpopulation it belongs to happens to be very different from all other subpopulations (like the Arctic fox or Red wolf).  In figure 4, we use $k=1$, which means that a sample only needs to be close(st) to one subpopulation from its true class in order to be correctly classified. We use $k=1$ for simplicity in Figure 4, and discuss the choice of $k$ in depth in Sec 5.4.
>
> 2. Thank you for pointing out our experiments warrant further investigation of finer-grained datasets. Based on your suggestion we ran extra experiments on 9 new datasets, which are nearly all fine-grained. Fortunately we find that our method still improves against all baselines consistently over datasets, with an average improvement of +1.3% over each baseline when using CLIP (see table A). Nonetheless, we will add a limitations section to discuss when our method may not be best suited (e.g. super fine-grained tasks with precise class names), at your suggestion.
>
> 3. In practice, hierarchy or attribute labels are not available. Thus, one must infer them. Doing so manually is costly, and even prohibitively so when the number of classes grows, or when you consider as many axes of variation as we do. In contrast, our method is fully automatic, allowing for the attribution of dozens of classes along many axes of variation to be done in just tens of minutes. Specifically, we timed how long it took to generate LLM responses to all queries in our method over multiple datasets, and found that **on one GPU (no parallelism), our attribution takes roughly 18.5 seconds per class**.  Thus, attributing datasets with 30 classes takes only about 10 minutes, and this number can be sped up by parallelizing. Crucially, this is done only once per classification task. Therefore, the computational cost per test image to classify is nearly equivalent to standard zero-shot classification.
>
> Feel free to comment if there are any lingering questions. If you feel that we have addressed all your concerns (namely that our method is too costly computationally or that it may not work for finer grained datasets), we would be very grateful if you could increase your score. Thank you!

---

### Official Review · Reviewer_KwVV · 2023-11-01

**Soundness:** 3 good
**Presentation:** 2 fair
**Contribution:** 1 poor
**Rating:** 3
**Confidence:** 4

**Summary:**

The paper attempts to alleviate the issue of single class names being used in image classification where those classes can be in fact, broad and diverse - in many possible aspects, like state, appearance, sub-groups/species, etc.
The authors argue that models do not have a mechanism for representing diversity within classes, and that models suffer from having to associate concepts/objects of potentially many subclasses or forms of objects in different state, under a single class.

To address this limitation of the models, the paper proposes a method that relies on querying an LLM for additional texts that could describe different variants of a class. Queries include prompting for possible attributes, subclasses, etc. (e.g. “pear” --> “whole pear”, “pear slices”; “wolf” --> “gray wolf”, “red wolf”, etc.).
Then, the authors classify among all possible generated additional classes, averaging predictions from the selected number of top subclasses (e.g. red wolf) to the original base class (e.g. wolf). This way, they hope to better capture some form of granularity or diversity within each class.

The proposed method is relatively similar to CHiLS (Novack et al. (2023)) not specific to hierarchies, however, but considers more possible types of “subclasses” or extended “classes” instead.

The paper contains experiments of the proposed method against baselines, such as using original classnames, and other relevant models, on a number of datasets that contain concepts that within classes are either hierarchical or appear in different states.

**Strengths:**

- (S1) The paper contains experiments on relatively many datasets of different kinds. The datasets used cover different types of structure and relations between classes: hierarchies, classes with different states and attributes. That gives a better understanding of how the model’s performance in wider range of scenarios. Although see W5

- (S2) From the technical point of view, the work has a sound and valid motivation (single class names as labels problematic for within class diversity)

- (S3) The approach proposed in the paper is technically simple and sound, does not seem to require modest extra computational resources. Although see W3.

**Weaknesses:**

- (W1) The performance improvement from the proposed approach is far from substantial. In many cases, the performance is almost equivalent to WaffleCLIP, which uses completely random text sequences.

- (W2) The motivation of the paper might not have much practical significance and the problem addressed appears to be somewhat artificial.
    The underlying issue behind the paper’s motivation seems mostly related to how classes in those datasets are constructured/selected, their granularity, structure, and relations between them.
    Whether e.g. Big Ben is a clock, a building, or a tower, basically depends on the problem underlying problem that one intends to solve. Many datasets are not made to solve any practical problem but to facilitate many types of research in general. Therefore, the classes in those datasets are defined in a way that might be very broad, capture many possible sub-categories, or the granularity of which is not practically usable. Using an example from the paper, classifying an “arctic fox” as a “fox” might marginally improve the accuracy numbers but is not necessarily a better output. Whether it is depends on the underlying problem one intends to solve. Similarly, would it necessarily be better for a classifier to predict tomato as a vegetable, not a fruit? Because the biological classification of a tomato is a fruit (a type of berry).
    The within-class “diversity” that the paper attempts to capture seems to be mostly relevant for datasets where labels somewhat artificially capture many possible sub-categories just because they can technically be marked under the same name. But for any practical applications, the label space/names should be defined more meaningfully.
    Also, considering the point above (W1), given the difference in performance is only marginal between models, if that difference comes from the technical correctness on the labels (e.g. “arctic fox” classified as a “fox”) that might necessarily mean that the model is more useful in practice. Also, see W5.

- (W3) Despite the approach being simple from the technical aspects (see S3), the model is dependent on the accuracy and structure of the LLM’s outputs. This requires tailoring queries/prompts for a specific dataset or a set of datasets.  Potentially, they could require a lot of tuning. Even though the set of queries used in the paper is fixed, and appears to work on all datasets, these are queries/prompts that had to be tuned/selected to be somewhat “compatible” with all datasets.

- (W4) The qualitative analysis (Figure 5, Appendix A) seems to consist of selected samples and likely does not represent the model’s predictions across the whole dataset accurately.

- (W5) The method is evaluated only on datasets which (in this case explicitly) contain some forms of sub-populations, hierarchies, or significant differences across attributes. Although this is an important analysis, the question of whether the method is only usable in these kinds of datasets is open. Would the method still be usable for datasets that might, but not necessarily do contain (at least not explicitly) some form of sub-groups or diversity within classes (maybe ImageNet for example?). Or datasets where not much diversity is expected, e.g. StanfordCars dataset?

**Questions:**

- (Q1) How exactly are the “worst” %x classes selected? Are they the same across all models or are they selected individually for each model? For Figures 6 (right) and 7, are they re-selected for every point (adding attributes, changing $k$ or $\lambda$ or kept the same?

- (Q2) For the Breeds dataset, on which level of the hierarchy of the labels the model is trained on?

- (Q3) Is the image sample of a “red wolf (in Figure 4) indeed a red wolf? Doing a quick search I am not so convinced that is what a red wolf looks like. Could it be a misclassified dog, for example? Do all other samples look similar to this one?

---

> ### Author Response · Authors · 2023-11-17
> **Clear advantages vs WaffleCLIP; Underperfoming on diverse instances is real world problem with precedent within the community**
>
> tldr: Per your suggestion (thx!), we conducted new experiments on 9 finer-grained datasets, including ImageNet and 4 of its variants. We observe gains for our method consistent with those seen in our original submission, suggesting that our method is not tuned to the original dataset suite and the prompts we used can be effective on new datasets with no tuning required. We hope this provides ample evidence demonstrating our method's performance gains over existing baselines.
>
> We sincerely thank the reviewer for taking the time to really engage with our paper and offer so much feedback. We now address weaknesses and concerns one by one.
>
> (W1) First, we note that despite WaffleCLIP’s simplicity, it is a recent and strong baseline. In all settings we study (hierarchical, states, real-world geographic diversity), we improve accuracy over WaffleCLIP. In fact, aside from GeoDe where there are fewer errors to correct, we improve accuracy by at least 0.9% for each setting (tables 1,2). These consistent gains also hold when using a different VLM (BLIP-2), where we improve by at least 0.8% for every setting aside from GeoDe (tables 7,8). These gains are consistent on each of the 8 datasets as well, as shown in table 9. **Averaging over all datasets and both VLMs, we improve accuracy compared to WaffleCLIP by 1.4%, with a larger gain of 2.6% and 2.4% for the worst 20% of classes and subpopulations respectively (see table 6)**.  We stress that these latter metrics (worst class/subpopulation) more closely reflect the core problem we seek to tackle, and see the larger gains there as encouraging evidence that our method improves upon the limitation it was designed to address (i.e. underperformance on atypically appearing instances).
>
> Moreover, **WaffleCLIP offers no interpretability, while we offer faithful and fine-grained explanations for each inference, for free** (Sec. 5.2). Similarly, overall, the reason why WaffleCLIP is effective is not well understood. In contrast, we make principled arguments as to why our method should work, and we support these arguments with extensive empirical evidence, such as notable gains for the least performant classes and subpopulations, and precise ablations (Sec. 5.3, 5.4).
>
> In summary, while WaffleCLIP is surprisingly strong, our method is (i) consistently stronger, (ii) unambiguously more interpretable, and (iii) better understood with respect to why it works.
>
> (W2) Respectfully, we disagree that diversity within classes is an artificial problem. Even something as simple and well defined as a ‘pear’ can actually manifest in many visually distinct ways. Moreover, this is a problem with precedent, as there are numerous existing efforts within the AI community to ensure models work beyond typical instances. For example, the field of out-of-distribution (OOD) generalization focuses on improving performance where some aspects of test inputs, such as spurious correlations [1,2,3] or domains [4], are shifted compared to the training data. Many works have also focused on algorithmic fairness towards mitigating biases, which often consist of model’s underperforming on instances that are less common / atypical (e.g. due to belonging to a demographic group underrepresented in the training data) [5]. In fact, our worst subpopulation/class metrics are inspired by the OOD community (e.g. ‘worst group accuracy’ is standard), and DollarStreet and GeoDe come from the fairness community, as they show diversity in images (resulting in harmful discrepancies in performance) is a naturally arising **real-world** problem. Notably, these datasets contain everyday objects whose labels are intuitive and unambiguous (that is, the class names are in no way designed to artificially capture many sub-categories). Many other benchmarks also exist to measure how classifiers will perform when encountering atypical data [6,7], suggesting this is a problem the scientific community deems important.
>
> Nonetheless, we agree that it's important to verify that our method works generally, and not just for one set of attributed datasets curated to inspect accuracy along various axes of diversity. We explore this below.
>
> [1] Distributionally Robust Neural Networks for Group Shifts: On the Importance of Regularization for Worst-Case Generalization, https://arxiv.org/abs/1911.08731
> [2] WILDS: A Benchmark of in-the-Wild Distribution Shifts, https://arxiv.org/abs/2012.07421
> [3] Invariant Risk Minimization, https://arxiv.org/abs/1907.02893
> [4] In Search of Lost Domain Generalization, https://arxiv.org/abs/2007.01434
> [5] Gender Shades: Intersectional Accuracy Disparities in Commercial Gender Classification, https://proceedings.mlr.press/v81/buolamwini18a.html
> [6] The Many Faces of Robustness: A Critical Analysis of Out-of-Distribution Generalization, https://arxiv.org/abs/2006.16241
> [7] ObjectNet: A large-scale bias-controlled dataset for pushing the limits of object recognition models, https://openreview.net/forum?id=SkgnRNHgIS

---

> > ### Author Response · Authors · 2023-11-17
> > **Consistent gains on new, finer-grained datasets; LLM prompts are generalizable, debuggable, and empirically reliable**
> >
> > **(W2 continued)** To show our method provides benefits on datasets beyond those used in the initial submission of this paper, we conduct additional experiments on the following 9 datasets: ImageNet, ImageNet variants (v2, -R, -A, -Sketch), Food-101, Flowers-102, FGVC-Aircraft, and Oxford Pets. Nearly all of these datasets are somewhat fine-grained*, and as such, are less likely to have intra-class diversity than our original datasets. Still, we observe gains consistent with the results presented in the original draft, with our method nearly always achieving highest or second highest accuracy: Out of 18 (9 datasets x 2 VLMs) settings, **we achieve highest accuracy in 13/18 cases (72.2%) and are in the top-2 in 15/18 cases (83.3%)**. Namely, **we improve accuracy over every baseline, and often see larger gains for the least performant classes**. Table A shows this pattern as averaged over all datasets: averaging the improvement of our method compared to each baseline, **we observe +1.32% and +1.74% for accuracy overall and on the worst 20% of classes** respectively using CLIP. Table(s) B shows the full results by dataset. For a specific example, when using CLIP to perform ImageNet classification, our method improves upon all baselines, including **+1.5% and +4.7% gains over WaffleCLIP** in accuracy overall and for the worst 20% of classes respectively. We appreciate the reviewer’s suggestion to check non-attributed datasets.
> >
> > Lastly, we note that diversity can be easily overlooked, as atypical examples are more rare. Thus, while more descriptive classnames certainly would improve zero-shot performance, it can be challenging for a human to (i) consider all the (many, at times unexpected) ways diversity may exist within each class and (ii) distill these into a single classname that is effective for all the subpopulations. Our method leverages an LLM with multiple queries to perform step (i) across all classes, and introduces a novel consolidation scheme that circumvents the dilemma of step (ii).
> >
> > *ImageNet (and its variants) contains 120 out of 1000 categories dedicated only to various dog breeds; Flowers-102 has many highly similar classes; Oxford Pets only has different breeds of dogs and cats; the FG in FGVC-Aircraft stands for fine-grained.
> >
> > **(W3)** The results presented in Tables A&B show that **the same set of queries we proposed can be used effectively on other datasets without tuning**. We note that we did not tune our queries to our original dataset suite (i.e. we avoided finding an optimal subset of queries or continuously adding new ones), and we most certainly did not tune them to the 9 new datasets, as these were selected after the method was fixed. Similarly, we did not tune our dataset suite to our method: we fixed our original datasets before obtaining results, and we do so again now (i.e. we did not remove any datasets where results were unfavorable to our method, like Oxford Pets). We strongly support transparency, and will release all code upon acceptance. In the same spirit, we’ll add a limitations section to discuss when our method may not be best suited (e.g. super fine-grained tasks with precise class names).
> >
> > We also concur that the LLM could be a source of error. However, unlike most components in an ML pipeline, the **LLM provides interpretable outputs, making debugging far easier**. Moreover, **our flexible consolidation scheme offers a kind of robustness to irrelevant LLM outputs**: Recall, only the similarity to a small number of subpopulations per class contribute to each logit. Thus, irrelevant (i.e. not appearing in the data) subpopulations are effectively ignored and do not affect the logit. This aspect of our work is what enables our method’s improved scaling as more attributes are incorporated: while existing methods saturate (averaging consolidation of Vanilla, DCLIP, WaffleCLIP) or deteriorate (CHiLS consolidation), our method continues to improve as more attributes are added (Section 5.3). Moreover, **LLM outputs are overwhelmingly accurate**, based on the following new small scale human validation: per trial, we randomly select a dataset, class within the dataset, and LLM query, and then manually check the fraction of responses that are appropriate. Out of 300 checked outputs, none are found to be incorrect, and only 8 are found to be uninformative. Namely, some responses to the descriptors query (actually from DCLIP) include categories as opposed to specific descriptors (e.g. ‘size and shape’ is given for ‘dog’ instead of ‘small’). Since these are merely uninformative and not incorrect, it is unlikely that our method will be damaged by them, as they’ll simply be ignored. Further, the rate of these responses is very low (2.7%). Thus, we believe the potential for erroneous LLM responses minimally affects our method. Nonetheless, we find automatic verification of LLM responses to be an interesting avenue of future research, and thank the reviewer for their comment.

---

> > > ### Author Response · Authors · 2023-11-17
> > > **Details and Results from evaluation on 9 new finer-grained datasets**
> > >
> > > We now present experimental results for nine additional datasets, along with citations for these datasets. These datasets are more fine-grained and do not contain attribute labels. The latter means that we cannot compute accuracy for worst subpopulations, as subpopulations cannot be defined without attribute labels. Table A presents a summarized view, showing on average how much we improve over each baseline. Complete results can be seen in Table(s) B.
> > >
> > > Table A: Average gains of our method compared to four baselines on 9 new (finer-grained) datasets. Accuracy overall and for the worst 20% of classes reported, using both CLIP ViT-B/16 and BLIP-2.
> > > | |CLIP| |BLIP-2| |
> > > |:----|:----|:----|:----|:----|
> > > |Gains over...|Accuracy|Worst 20%|Accuracy|Worst 20%|
> > > |Vanilla|+1.81|+1.51|+0.55|+0.70|
> > > |DClip|+0.91|+1.77|+0.69|+1.67|
> > > |Waffle|+1.50|+2.72|+0.28|+0.56|
> > > |CHiLS|+1.06*|+0.97*|+0.27|+0.08|
> > > |Average |+1.32|+1.74|+0.45|+0.75|
> > >
> > > *When computing average gain over CHiLS for CLIP, we exclude ImageNet related datasets because **CHiLS fails catastrophically**. We conjecture the issue arises due to the high number of classes (1k), and even larger set of subclasses (about 10k). Each logit in CHiLS is the product of a softmax output over 1k options and a softmax output over about 11k options. Furthermore, because CLIP similarities usually fall within a small range (about 0.1-0.3), the difference in final logits may be so small that noise from rounding errors dominates the signal. CHiLS does not fail on ImageNet when using BLIP-2, suggesting our implementation is correct, and the problem arises due to small differences in CLIP similarities. One could likely fix this problem by changing the temperature of the softmax, but we opt to faithfully follow the original method. Recall CHiLS was only intended for datasets with clear hierarchy, whereas our method is more versatile and generally effective: we work well at ImageNet scale, and even without including results for the five ImageNet related datasets where our method is strong while CHiLS achieves near zero accuracy, we still beat CHiLS by over 1% averaged across the other datasets.
> > >
> > > Table(s) B: Performance over each of the 9 additional datasets. Best method bolded, second best underlined.
> > > Table B1. VLM: CLIP ViT-B/16, Metric: Accuracy
> > > | |ImageNet|v2|-A|-R|Sketch|Food|Flowers|Aircraft|Pets|
> > > |:----|:----|:----|:----|:----|:----|:----|:----|:----|:----|
> > > |Vanilla|68.48|61.98|30.16|59.24|48.37|88.35|66.09|31.26|**92.72**|
> > > |DClip|*68.85*|*62.37*|*31.35*|60.04|48.54|88.05|**70.69**|*32.67*|92.23|
> > > |Waffle|68.44|62.15|31.09|*61.17*|*48.58*|88.09|66.87|31.05|92.03|
> > > |CHiLS|0.11|0.1|0|0|0.11|*88.59*|67.49|34.2|92.12|
> > > |Ours|**69.94**|**63.32**|**32.19**|**61.49**|**49.38**|**89.06**|**70.69**|**34.62**|*92.26*|
> > >
> > > Table B2. VLM: BLIP-2, Metric: Accuracy
> > > | |ImageNet|v2|-A|-R|Sketch|Food|Flowers|Aircraft|Pets|
> > > |:----|:----|:----|:----|:----|:----|:----|:----|:----|:----|
> > > |Vanilla|55.68|51.26|26.53|63.06|55.35|83.73|53.05|25.65|**65.33**|
> > > |DClip|56.15|51.44|*26.6*|62.53|54.79|84.02|**54.12**|26.43|62.36|
> > > |Waffle|*57.1*|*52.11*|**26.83**|*64.78*|*55.88*|83.61|52.03|**27.00**|62.78|
> > > |CHiLS|56.26|51.56|25.63|63.43|55.54|84.29|53.55|*26.94*|*64.95*|
> > > |Ours|**57.28**|**52.27**|26.25|**65.13**|**56.21**|**84.79**|54.03|25.23|63.42|
> > >
> > > Table B3. VLM: CLIP ViT-B/16. Metric: Accuracy on worst 20% of classes.
> > > | |ImageNet|v2|-A|-R|Sketch|Food|Flowers|Aircraft|Pets|
> > > |:----|:----|:----|:----|:----|:----|:----|:----|:----|:----|
> > > |Vanilla|34.42|*25.75*|*7.47*|*29.27*|*9.19*|73.58|2.08|0|**78.84**|
> > > |DClip|*34.58*|25.65|6.24|28.9|8.99|72.92|*3.44*|0|*77.5*|
> > > |Waffle|32.62|23|5.79|28.18|8.93|73.36|2.14|0|75.65|
> > > |CHiLS|0|0|0|0|0|*75.36*|2.09|0|76.92|
> > > |Ours|**37.3**|**27.6**|**7.98**|**33.74**|**9.33**|**76.18**|**4.4**|**0.25**|77.4|
> > >
> > > Table B4. VLM: BLIP-2. Metric: Accuracy on worst 20% of classes.
> > > | |ImageNet|v2|-A|-R|Sketch|Food|Flowers|Aircraft|Pets|
> > > |:----|:----|:----|:----|:----|:----|:----|:----|:----|:----|
> > > |Vanilla|6.74|5.4|**3.39**|24.12|3.99|57.36|0|0|**27.52**|
> > > |DClip|7.62|*6.55*|2.8|22.83|4.2|58.52|0|0|17.34|
> > > |Waffle|**8.22**|5.98|*3.25*|*24.16*|**5.14**|57.21|0|0|25.84|
> > > |CHiLS|8.05|6.25|2.9|24.02|*4.87*|*60.9*|0|0|*27.13*|
> > > |Ours|*8.19*|**6.9**|2.87|**25.65**|4.53|**61.44**|0|0|25.28|
> > >
> > > Test split used for all datasets unless otherwise mentioned.
> > > - ImageNet: https://www.image-net.org/, we use ‘val’ split
> > > - ImageNet v2: https://github.com/modestyachts/ImageNetV2, we used ‘matched frequency’ split
> > > - ImageNet-A: https://github.com/hendrycks/natural-adv-examples
> > > - ImageNet-R: https://github.com/hendrycks/imagenet-r
> > > - ImageNet-Sketch: https://github.com/HaohanWang/ImageNet-Sketch
> > > - Food-101: https://data.vision.ee.ethz.ch/cvl/datasets_extra/food-101/
> > > - Flowers-102: https://www.robots.ox.ac.uk/~vgg/data/flowers/102/
> > > - FGVC Aircraft: https://www.robots.ox.ac.uk/~vgg/data/fgvc-aircraft/, we used ‘test’ split for ‘manufacturers’ task
> > > - Oxford Pets: https://www.robots.ox.ac.uk/~vgg/data/pets/

---

> > > > ### Author Response · Authors · 2023-11-17
> > > >
> > > > **(W4)** Like in any qualitative validation, we selected a few interesting examples. However, we tried our best to avoid extreme cherry picking by only inspecting 100 random instances where our method corrected mistakes of the standard method, and picking out of those. **We would be happy to include a purely random sample of similar inferences** in the camera ready revision draft to be fully transparent. Again, we also will release all code upon acceptance. Further, since **the LLM we utilize is open source** (unlike CHiLS and DCLIP), **all our experiments will be fully reproducible by anyone for free**.
> > > >
> > > > **(W5)** This is a great and very valid point. We genuinely believe diversity is everywhere, whether or not its presence is noted. We take your suggestion in stride and obtain results over 9 new datasets (detailed above), finding gains consistent with our original results. We find that **seeing our method also improve performance for finer-grained datasets is a promising indicator that intra-class diversity may arise even when we do not expect it, and consequently that our method provides meaningful gains in settings beyond those evaluated in the original submission**. If there are any more datasets you would like us to try, we would be happy to give them a shot (time permitting).
> > > > _____
> > > > We now provide answers to your questions as well.
> > > >
> > > > **(Q1)** The worst classes and subpopulations are chosen separately for each run, as is common practice in fairness literature (e.g. ‘worst group accuracy’). Nonetheless, very frequently, they correspond to the same classes / subpopulations.
> > > >
> > > > **(Q2)** The level of ImageNet hierarchy varies for the four different Breeds datasets. Two datasets (Living17, Nonliving26) are at level 5, and the other two are at levels 3 and 4. We note that these four datasets were heavily relied upon in CHiLS; we include them to compare our method to the CHiLS baseline at its strongest.
> > > >
> > > > **(Q3)** That image is indeed a red wolf. It is labeled as such in ImageNet, and its ears are more similar to a wolf’s than a dog. Other samples in the dataset indeed look similar, though perhaps slightly more wolf-like (often evident when a side profile of the wolf is visible). The fact that it arguably looks more like a dog than a wolf precisely illustrates a motivation to our method: some subpopulations within a class may be more similar to most instances of another class.
> > > >
> > > > Thank you very much for reading all the way to the end of our long rebuttal! Please leave any more questions or comments below; we’d be more than happy to answer. If you feel we have addressed your concerns and adequately shown our method has clear novelty and advantages over existing art, we’d greatly appreciate it if you could increase your score to an accept.

---

> > > > > ### Author Response · Authors · 2023-11-21
> > > > >
> > > > > Hi, just a friendly reminder about our rebuttal above. We appreciate the time you've spent reviewing our paper thus far, and would be very grateful if you could consider our responses before the **discussion period ends tomorrow**.
> > > > >
> > > > > The main update from our rebuttal is that **we tested on 9 new fine grained datasets** and found our method consistently improves over strong baselines in these settings, showing that **our approach works (a) without tuning (b) in settings where intra-class diversity is not guaranteed to be present**, suggesting that the problem we tackle is practical and our method is effective.
> > > > >
> > > > > Thank you!

---

> > > > > > ### Comment · Reviewer_KwVV · 2023-11-22
> > > > > > **Response to the rebuttal**
> > > > > >
> > > > > > I want to thank the authors for the extensive response!
> > > > > >
> > > > > > Unfortunately, I do not think the weaknesses (W1), and (W2) I described have been effectively addressed.
> > > > > >
> > > > > > WaffleCLIP is a dummy baseline that uses random texts.
> > > > > > I would consider it as a baseline of a form of a sanity check of proposed alternative methods, not some SOTA model that one tries to outperform by a percent or two. Because what would that tell us?
> > > > > >
> > > > > > I apologize if I was not precise enough with (W2) because my comment seems to be interpreted not exactly as I intended it to be - I did not mean to say that within-class diversity itself is not important or an artificial problem.
> > > > > > Rather, I meant to question using a setting of image classification datasets where class definitions or granularity is somewhat artificial and not defined to solve any particular underlying problem.
> > > > > >
> > > > > > I elaborated more in my initial comment, providing examples, but what is referred to in the paper as within-class "diversity" is in many datasets an artifact of how labels are selected.
> > > > > > Objects/concepts share the same class only because they technically could be described with the same word or because of biological relations (which again, might be very different from how humans would describe them - tomato being a cherry), without considering the underlying problem such image classification would aim to solve.

---

> > > > > > > ### Author Response · Authors · 2023-11-22
> > > > > > >
> > > > > > > Thanks for your reply!
> > > > > > >
> > > > > > > **For W2, we call attention to our new results**: on a brand new set of datasets, which, notably, are finer-grained and do not contain ‘artificial’ granularity, our method still proves to be consistently stronger than all baselines. We’d appreciate it if these results could be acknowledged, as we believe they address the concern you raised (i.e. if our method works outside of the original setting of evaluation; it does).
> > > > > > >
> > > > > > > Also, DollarStreet and GeoDe are real-world fairness benchmarks, whose **labels are the common names** for household objects. Thus, it is unreasonable to say the diversity is only an artifact of how the labels are selected, as labels are selected in the most natural way.
> > > > > > >
> > > > > > > As for W1, despite WaffleCLIP’s simplicity, it is surprisingly strong, often beating other SOTA methods (DCLIP, CHiLS*). We think WaffleCLIP’s success tells us that accuracy can be improved by considering multiple vectors instead of just the single (potentially noisy) vector obtained from embedding the classname alone. However, importantly, WaffleCLIP offers no clear answer as to what its gains signify and where they come from. In contrast, we detail how our method provides gains (e.g. on atypical subpopulations) and provide extensive empirical evidence supporting our claim (e.g. numerous examples, precise ablations, and careful metric selection). Further, we show and explain how our novel consolidation method yields better scaling as new attributes are incorporated. Lastly, our method provides interpretability, while WaffleCLIP has none.
> > > > > > >
> > > > > > > We believe that our consistent 1-2% gains over a total of 18 datasets indicate that our improvements over WaffleCLIP (and other baselines) are beyond just noise, especially since we are operating in a zero-shot training-free setting, where gains are always limited by the fact that the underlying model is entirely frozen.
> > > > > > >
> > > > > > > In summary, WaffleCLIP offers consistent, non-trivial accuracy improvements over the vanilla baseline, even if it appears like a ‘dummy’ baseline. Similarly, our method offers consistent, non-trivial improvements over WaffleCLIP (and all others) in accuracy, but importantly, also in having enhanced interpretability and a well understood source of gains. We hope this elucidates the clear advantages of our approach over existing art.
> > > > > > >
> > > > > > > *We do our best to compare to the strongest existing baselines. If the reviewer believes another method better represents SOTA, we'd be happy to compare our method against it.

---

> > > > > > > > ### Comment · Reviewer_KwVV · 2023-11-23
> > > > > > > > **Final confirmation**
> > > > > > > >
> > > > > > > > I just want to confirm (to both the Authors, AC, and other Reviewers) that I have indeed seen the added results.
> > > > > > > >
> > > > > > > > However, I do not believe that the above comments are to the point concerning the weaknesses I describe.
> > > > > > > >
> > > > > > > > Additionally, I find the above characterization of WaffleCLIP and the meaning of it as a baseline, misleading.
> > > > > > > > Even the WaffleCLIP paper refers to their method as a "sanity-check" for other methods.
> > > > > > > > The authors seem to miss the point of comparing to this kind of dummy/sanity-check baselines.

---

### Author Response · Authors · 2023-11-18
**Thank you to all reviewers, some general comments**

Thank you to all reviewers for their insightful comments. We appreciate that the reviewers found our method to be intuitive (Reviewer NLHE), having a sound and valid motivation (Reviewer KwVV), and effective (Reviewer 3g4D). In our rebuttal, we address all raised concerns, including but not limited to showing that:
- our results from the original submission show gains over multiple strong baselines in numerous settings
- new **additional results** show our method can generalize out of the box to new settings, and still provide consistent gains
- our generation of LLM outputs (i) does not require prompt tuning (ii) is quick and (iii) yields overwhelming accurate outputs (via a small-scale human validation study)
-the limitation we tackle (model underperformance on atypical instances) is **real and practical**, evident by the many related efforts of the OOD generalization and fairness communities

We highlight that **we evaluate our method on 9 new finer grained dataset**, including ImageNet and four variants, and find **our method offers consistent gains (+1.32% and +1.74% average gains in accuracy overall and for the worst 20% of classes using CLIP)**, mirroring the results in our main text, and thus showing that our method is effective in general settings without needing any tuning. We hope that this point, in addition to the significantly enhanced interpretability, clear motivation, and well understood sources of gains for our method, provides ample evidence demonstrating the advantages of our approach over existing art.

---

### Meta-Review · Area_Chair_WSKv · 2023-12-06

**Metareview:**

This paper proposes a method to encode and account for diversity within a class using inferred attributes for zero-shot classification.  Three reviewers provided ratings of 3, 6, 5.  Positive points include sound motivation, and simple and effective approach.  Negative points include limited performance improvements, question on the work's practical significance, computation cost, and missing baselines/datasets.  The rebuttal (including the new experiments) and authors' messages to the ACs addressed several of these concerns, including practical significance, computation cost, and missing baselines/datasets.  There was an extensive discussion regarding limited performance improvement over closely related baselines, especially WaffleCLIP.  In the end, two reviewers still had concerns regarding this point.  Overall, this is a borderline paper with no strong advocate for accepting the paper.  The paper, rebuttal, discussion, and author messages were carefully discussed among the ACs, and, unfortunately, the ACs agree that the paper as it stands does not meet the bar for ICLR acceptance.  The ACs would like to encourage the authors to improve the paper and resubmit to another conference.

**Justification For Why Not Higher Score:**

The paper is not ready for publication based on the points mentioned in the meta review.

**Justification For Why Not Lower Score:**

N/A

---

### Decision · Program_Chairs · 2024-01-16

Reject